# Geometric Flow Grounding: A Unified Manifold Decoupling Framework for Dynamics Discovery and Verification

Chang Yu [* 1]  Yuxuan Luo [* 2]  Yixuan Du [3]  Yuqing Zhou [1]  Siyuan Li [1 2]  Jingbo Zhou [1 2]  Jiawei Jiang [4]
Zhen Lei [5 6 7]  Stan Z. Li [1]

## Abstract

Modeling complex dynamics from observational data is fundamental to scientific discovery and artificial intelligence. However, existing approaches are often plagued by the entanglement of static state representations and instantaneous motion, leading to accumulated errors and off-manifold hallucinations where predicted trajectories violate intrinsic geometric constraints. To address this, we propose Geometric Flow Grounding, a unified framework that enforces dynamic evolution strictly along the tangent bundle of the learned data manifold via a differentiable Neural Tangent Projection Layer. By geometrically decoupling state representation from tangential dynamics, our method generalizes across diverse data regimes. In scientific discovery, GFG reduces numerical aliasing and improves long-horizon stability in sparse dynamical systems, while recovering interpretable gene regulatory motifs from single-cell data. For trustworthy AI, the projection residual provides a zero-shot metric for deepfake video detection by revealing inconsistencies with the implicit flow of pre-trained world models. Our results establish manifold-constrained projection as a universal operator for both discovering natural laws and verifying synthetic content. Code will be available at https://github.com/yuchang97/GFG-public

[1]Westlake University, Hangzhou, China [2]Zhejiang University, Hangzhou, China [3]Tianjin University, Tianjin, China [4]The College of Computer Science and Technology, Zhejiang University of Technology, Hangzhou, China [5]MAIS, Institute of Automation, Chinese Academy of Sciences, Beijing, China [6]School of Artificial Intelligence, University of Chinese Academy of Sciences, Beijing, China [7]CAIR, Hong Kong Institute of Science and Innovation, Chinese Academy of Sciences, HongKong, China. Correspondence to: Zhen Lei <zhen.lei@ia.ac.cn>, Stan Z. Li <Stan.ZQ.Li@westlake.edu.cn>.

*Proceedings of the $43^{rd}$ International Conference on Machine Learning*, Seoul, South Korea. PMLR 306, 2026. Copyright 2026 by the author(s).

## 1. Introduction

Learning continuous dynamics from high-dimensional observations is a core problem shared by scientific modeling and modern generative AI (Neklyudov et al., 2023; Chen et al., 2021). This problem spans diverse fields, ranging from recovering physical laws from sparse data (Brunton et al., 2016; Anvari et al., 2025) and inferring cellular transitions from static snapshots (Bergen et al., 2020; Gayoso et al., 2024; Cui et al., 2024), to synthesizing consistent motion in video generation (Wiedemer et al., 2025). Despite their differences, these tasks share a common goal: learning a vector field that produces plausible trajectories beyond the observed data.

Current approaches, such as Neural ODEs and continuous normalizing flows, typically parameterize vector fields directly in the ambient Euclidean space, optimizing for reconstruction or likelihood without explicit geometric constraints. However, treating the high-dimensional latent space as flat ignores the intrinsic curvature of the data support. This geometric agnosticism leads to two critical forms of off-manifold error (see Figure 1 (a)). First, *smoothing drift* occurs when the model averages conflicting velocity signals (e.g., $v_a$ and $v_b$) in a local neighborhood, resulting in an interpolated vector $v_c$ that points orthogonal to the manifold surface. Second, *ambient rollout drift* can arise during numerical integration: even if the instantaneous velocity is tangential at the current point, a discrete step taken directly in the ambient space may leave a curved manifold unless the state is re-parameterized or corrected. Without geometric control, such normal deviations can accumulate and drive trajectories into unrealistic regions of the state space.

We argue that physical dynamics should not be modeled as arbitrary ambient-space motion, but should align with the local tangent structure of the underlying data manifold. This principle aligns with foundational works in geometric deep learning (Bronstein et al., 2017) and has inspired recent advances like Neural Manifold ODEs (Lou et al., 2020) and Riemannian Flow Matching (Chen & Lipman, 2024). While these methods provide important geometric reference points, they often assume known or analytically specified manifold geometry. In high-dimensional observational domains such

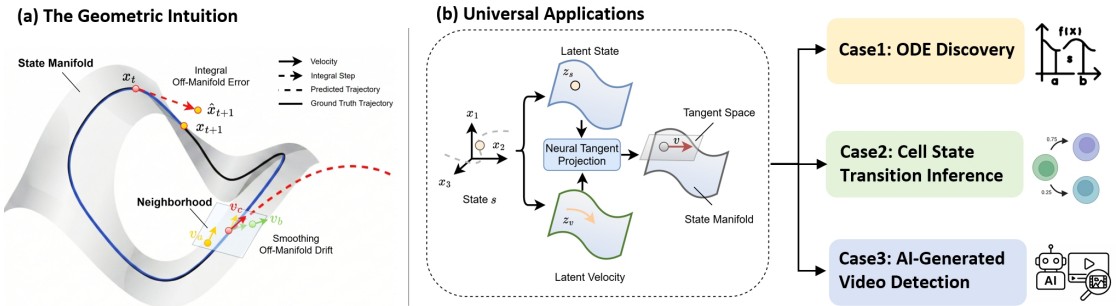

*Figure 1.* **Geometric Flow Grounding: Intuition and Applications. (a) The Geometric Intuition.** Standard dynamic models parameterize vector fields in the flat ambient space, leading to two critical failures: (1) *Integration Error*, where discrete numerical steps ($x_t \to \hat{x}_{t+1}$) inevitably push the predicted trajectory (red dashed line) off the manifold; and (2) *Smoothing Drift*, where the model averages conflicting velocities in a neighborhood ($v_a, v_b$) into an invalid vector $v_c$ orthogonal to the surface. **(b) Universal Applications.** To resolve this, our framework employs a *Neural Tangent Projection* mechanism (center) that strictly grounds latent velocities ($z_v$) onto the tangent space of the state manifold ($z_s$). This geometric constraint serves as a unified foundation for three distinct tasks.

as gene expression and video, the data manifold is rarely known a priori. Chart-dependent operations or divergence estimation can introduce nontrivial overhead.

To operationalize geometric constraints efficiently without these limitations, we propose Geometric Flow Grounding (GFG). GFG is a unified framework that enforces this constraint through two synergistic mechanisms: *Neural Tangent Projection* and *Velocity Primitive Quantization*. To eliminate integration drift, our projection mechanism leverages automatic differentiation to implement a strict geometric constraint. Specifically, we utilize the Jacobian-Vector Product (JVP) of the generator to explicitly map predicted velocities onto the local tangent space, ensuring trajectories remain mathematically grounded on the manifold surface. Simultaneously, to resolve smoothing drift caused by conflicting velocity signals, we introduce a learnable codebook of velocity primitives. By quantizing continuous dynamics into discrete, reusable tangent basis vectors, we force the model to select distinct dynamic modes rather than regressing to an off-manifold mean.

We validate Geometric Flow Grounding across three distinct regimes, each chosen to isolate and address a fundamental challenge in dynamic modeling (Figure 1 (b)). First, in sparse dynamical system identification, we examine whether strict tangent constraints can mitigate the long-term numerical divergence typical of unconstrained baselines. Second, in single-cell biology, we investigate the capacity of our disentangled representations to resolve the averaging dilemma inherent in complex lineages, aiming to recover sharp branching points where standard smoothing methods fail. Finally, to demonstrate the universality of these geometric principles, we extend our analysis to AI-generated video detection. This setting serves to verify that adherence to the underlying data manifold functions not only as a constraint for scientific discovery but also as a rigorous metric for distinguishing physical reality from synthetic hallucinations. Our main contributions are summarized as follows:

- *Unified Geometric Framework.* We introduce Neural Tangent Projection, a differentiable mechanism utilizing matrix-free Jacobian-Vector Products (JVP). By strictly grounding latent dynamics on the data manifold, this approach eliminates integration error and prevents the numerical divergence inherent in unconstrained baselines without the computational cost of explicit coordinate charts.

- *Disentangled Dynamics via Quantization.* We propose a dual-stream architecture that explicitly separates topological states from discrete velocity primitives. This quantization-based design resolves the smoothing drift caused by conflicting flows, enabling the unsupervised discovery of interpretable laws and sharp branching points in complex biological lineages.

- *Training-free Physical Verification.* We establish the geometric projection residual as a rigorous, training-free metric for consistency auditing. By repurposing our constraint as a physics critic, the framework effectively detects off-manifold artifacts in generative video, exposing deepfakes that violate the implicit laws of pre-trained world models.

## 2. Preliminaries

**Problem Formulation.** We consider the general problem of learning the governing dynamics of a complex system from discrete snapshots. Let $\mathcal{X} \subseteq \mathbb{R}^D$ denote the high-dimensional observation space (e.g., pixel space for videos or gene expression space for cells). We are given a dataset of state observations $\mathcal{D} = \{x_t^{(i)}\}_{i,t}$, sampled from continuous-time trajectories.

Mathematically, we assume the evolution of the system is governed by a time-invariant vector field $v^* : \mathcal{X} \to \mathbb{R}^D$.

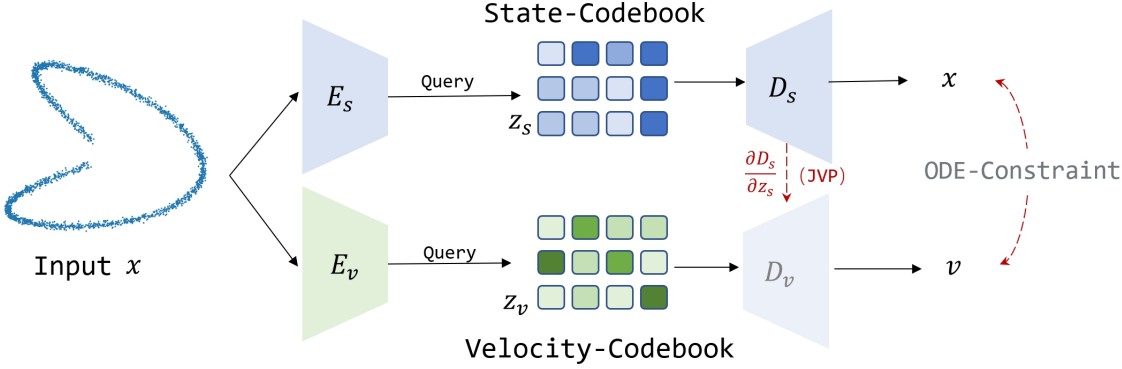

*Figure 2.* Overview of the Geometric Flow Grounding (GFG) Framework. GFG employs a dual-stream architecture that explicitly disentangles spatial topology from temporal dynamics. (1) The *State Stream* (top, blue) constructs the manifold support by quantizing observations $x$ into discrete topological codes $z_s$. (2) The *Dynamics Stream* (bottom, green) combats *smoothing drift* by mapping evolution trends to a codebook of reusable dynamics primitives ($z_v$), forcing the selection of distinct dynamic modes to resolve conflicting flows. (3) Crucially, the *Neural Tangent Projection* (red dashed arrow) grounds these latent primitives onto the tangent bundle. By computing the Jacobian-Vector Product (JVP) of the state generator $D_s$, GFG removes the normal component of the velocity and mitigates off-manifold drift through latent-space integration and decoding.

For any state $x_t$, the instantaneous trend is described as:

$$\frac{dx(t)}{dt} = v^*(x(t)). \tag{1}$$

Our primary objective is to learn a parameterized estimator $v_\theta : \mathcal{X} \to \mathbb{R}^D$ that approximates the ground truth field $v^*$, allowing us to reconstruct trajectories via numerical integration $x_{t+\tau} = x_t + \int_t^{t+\tau} v_\theta(x_s)ds$ or to verify the plausibility of observed transitions.

**The Manifold Hypothesis.** A fundamental challenge in solving Eq. (1) directly in high-dimensional spaces is the sparsity of data relative to the volume of $\mathcal{X}$. To address this, we adopt the standard *Manifold Hypothesis*, which posits that the effective degrees of freedom of the system are much lower than the ambient dimension $D$. Formally, we assume the data support lies on a lower-dimensional Riemannian manifold $\mathcal{M} \subset \mathcal{X}$. In the context of deep generative modeling, this manifold is typically defined as the image of a smooth generator function $G$:

$$\mathcal{M} = \{G(z) \mid z \in \mathcal{Z}\}, \tag{2}$$

where $\mathcal{Z} \subseteq \mathbb{R}^d$ ($d \ll D$) represents the intrinsic latent space, and $G : \mathcal{Z} \to \mathcal{X}$ is a non-linear mapping used for decoding.

Under this setup, the problem transforms into learning dynamics that are consistent with the geometry implied by $G$. While standard approaches often model $v_\theta$ freely in the ambient space $\mathcal{X}$, this neglects the structural constraint that valid evolution must conform to the topology of $\mathcal{M}$.

## 3. Method

In this paper, we propose Geometric Flow Grounding (GFG), a unified framework designed to learn continuous dy-

namics strictly confined to the underlying data manifold. By explicitly projecting predicted velocity motifs onto the tangent bundle via a differentiable Neural Tangent Projection mechanism, GFG eliminates off-manifold hallucinations by construction. In this section, we first provide the theoretical justification for this constraint (Sec. 3.1), followed by the proposed dual-stream architecture (Sec. 3.2) and the geometric projection implementation (Sec. 3.3). We demonstrate the universality of this framework across diverse domains, ranging from equation discovery and single-cell RNA velocity to deepfake detection in video generation.

### 3.1. Theoretical Motivation: Geometric Grounding

Building on the learned manifold hypothesis (Sec. 2), we formalize why tangent-space grounding is beneficial for learning dynamics from high-dimensional observations. This perspective follows recent discussions of deep generative models under the manifold hypothesis (Loaiza-Ganem et al., 2024) and prior work showing that VAE-style decoders induce useful latent manifold geometries (Chen et al., 2020). Since the data manifold is not available a priori, our analysis is stated with respect to the learned manifold induced by the state decoder. Let $G_\theta : \mathcal{Z} \to \mathcal{X}$ denote the learned generator and let $J_\theta(z)$ be its Jacobian at latent state $z$.

**Assumption 3.1** (Learned manifold regularity). For encoded data points $z = E_s(x)$ with $x \in \mathcal{D}$, we assume: *(i) Local smoothness:* $G_\theta$ is locally $C^1$-differentiable, so that Jacobian-vector products are well defined. *(ii) Stable tangent space:* $J_\theta(z)$ has locally stable rank, with its nonzero singular values bounded away from zero and infinity. *(ii) Tangential dominance:* the dominant deterministic component of the dynamics is approximately tangent to the data support, while normal components correspond to noise, stochasticity, or manifold approximation error.

Under Assumption 3.1, the tangent space of the learned manifold is given by $\mathrm{Range}(J_\theta)$. Let $\mathbf{P}_{J_\theta}$ be the orthogonal projection onto this learned tangent space. The following result clarifies the role of tangent projection during inference and training.

**Theorem 3.2** (*Projection and Manifold Alignment*)**.** *Let $f^*$ be the ground-truth vector field, $\hat{f}_{\mathrm{amb}}$ be an unconstrained ambient estimator, and $\mathbf{P}_{J_\theta}$ be the orthogonal projection onto $\mathrm{Range}(J_\theta)$.*

*(1) Inference denoising. The projected estimator admits the decomposition*

$$\|\mathbf{P}_{J_\theta}\hat{f}_{\mathrm{amb}}-f^*\|_2^2 = \|\mathbf{P}_{J_\theta}(\hat{f}_{\mathrm{amb}}-f^*)\|_2^2+\|(I-\mathbf{P}_{J_\theta})f^*\|_2^2. \tag{3}$$

*Thus, tangent projection removes the component of the estimator error orthogonal to the learned tangent space, while the remaining residual is determined by the non-tangential component of the true dynamics. In the special case where $f^* \in \mathrm{Range}(J_\theta)$, this reduces to*

$$\|\mathbf{P}_{J_\theta}\hat{f}_{\mathrm{amb}} - f^*\|_2 \le \|\hat{f}_{\mathrm{amb}} - f^*\|_2. \tag{4}$$

*(2) Manifold alignment. For an observed transition $\Delta x$, the best tangent-space explanation is obtained by the least-squares projection*

$$\min_u \|\Delta x - J_\theta u\|_2^2 = \|(I - \mathbf{P}_{J_\theta})\Delta x\|_2^2. \tag{5}$$

*Therefore, minimizing this residual encourages the learned Jacobian, and hence the learned tangent space, to align with observed transitions.*

*Proof Sketch.* For (1), decompose $f^*$ into its tangent and normal components with respect to $\mathrm{Range}(J_\theta)$. Since $\mathbf{P}_{J_\theta}(\hat{f}_{\mathrm{amb}} - f^*)$ lies in the tangent space and $(I - \mathbf{P}_{J_\theta})f^*$ lies in the orthogonal normal space, Eq. (3) follows from the Pythagorean theorem. If $f^*$ is exactly tangent to the learned manifold, the normal residual vanishes, yielding Eq. (4). For (2), the constrained tangent fitting problem is a linear least-squares problem whose residual is exactly the orthogonal projection residual in Eq. (5). This shows that the physics consistency loss regularizes the learned geometry toward the observed dynamics, rather than applying a fixed tangent constraint post hoc. A full derivation and the imperfect-manifold error decomposition are provided in Appendix A.

### 3.2. Dual-Stream Disentangled Architecture

To mitigate the off-manifold deviations inherent in entangled modeling, we propose a Dual-Stream Architecture (Figure 2) that explicitly disentangles the representation of static topological identity from dynamic evolution trends. This design allows the model to learn a robust geometric support independent of complex dynamics. The framework comprises two parallel pathways:

*State Stream (Topology Learning).* The upper branch serves as the static manifold generator. The state encoder $E_s$ first projects high-dimensional observations $x \in \mathcal{X}$ into a latent space. To capture the discrete nature of underlying regimes (e.g., metastable states), we employ a topology codebook $\mathcal{C}_{topo} = \{e_k\}_{k=1}^K$. The continuous embedding is discretized to its nearest prototype $z_s \in \mathcal{C}_{topo}$. These discrete states are then mapped back by the decoder $D_s$ (acting as generator $G$) to reconstruct $\hat{x}$, thereby parameterizing the manifold support $\mathcal{M}$.

*Dynamics Stream (Primitive Discovery).* Parallel to the topology branch, the lower stream captures the system's evolution tendency. Sharing the same input as the state stream, the velocity encoder $E_v$ is specialized to extract kinetic cues rather than static identity. Analogous to the discrete topology, we posit that governing dynamics are composed of a finite set of intrinsic patterns (e.g., cell cycle phases or attractor basins). Consequently, instead of directly regressing unconstrained vectors, we introduce a dynamics primitive codebook $\mathcal{C}_{prim} = \{p_l\}_{l=1}^L$ to store these fundamental modes. The latent velocity $v_{lat} \in \mathbb{R}^d$ is formulated as a weighted combination of these primitives:

$$v_{lat} = \sum_{l=1}^L \alpha_l p_l, \quad \text{s.t.} \ \sum_{l=1}^L \alpha_l = 1, \tag{6}$$

where $\alpha_l$ denotes the attention weights predicted by $E_v$. This dictionary-based formulation aligns dynamic modeling with topological quantization, enhancing both robustness and interpretability.

However, the inferred primitive composition $v_{lat}$ resides strictly within the intrinsic latent space. To translate this abstract tendency back to the high-dimensional observation space while preserving the geometric constraints established in Sec. 3.1, we employ the neural tangent projection mechanism described below.

### 3.3. Neural Tangent Projection

A critical challenge in this dual-stream design is the decoding of the latent velocity $v_{lat}$. Simply training a separate, unconstrained decoder to regress the ambient velocity would decouple the dynamics from the topology, ignoring the local curvature captured by the State Stream. To bridge this gap and enforce the theoretical guarantees derived in Eq. 4, we introduce the neural tangent projection mechanism.

Mathematically, the tangent space $\mathcal{T}_{G(z)}\mathcal{M}$ is spanned by the columns of the Jacobian $J_G(z)$. Explicitly instantiating this high-dimensional Jacobian is computationally prohibitive. Instead, we leverage the property that the directional derivative of the generator along a latent vector $v_{lat}$ yields a vector strictly in the tangent space. We compute this efficiently using the Jacobian-Vector Product (JVP)

operator:

$$v_{tan} = \text{JVP}(G, z, v_{lat}) \triangleq \left. \frac{\partial}{\partial \epsilon} G(z + \epsilon v_{lat}) \right|_{\epsilon=0}. \quad (7)$$

As formally proven in Proposition A.1 (Appendix A), this operation maps the learned latent motifs $v_{lat}$ directly to a tangent velocity $v_{tan} \in \mathbb{R}^D$. Crucially, by construction, $v_{tan}$ has zero component in the normal space $\mathcal{N}_{G(z)}\mathcal{M}$, thereby ensuring that the generated dynamics are geometrically consistent and free from off-manifold hallucinations.

### 3.4. Optimization & Objectives

To ensure the learned manifold and dynamics are both geometrically accurate and physically meaningful, we propose a unified objective function. The total loss $\mathcal{L}_{total}$ is composed of a topological reconstruction term, which is universal across domains, and a generalized physical consistency term that is domain-specific:

$$\mathcal{L}_{total} = \mathcal{L}_{topo} + \lambda \cdot \mathcal{L}_{phy}. \quad (8)$$

The first term ensures the State Stream learns a high-fidelity geometric support. We employ the standard VQ-VAE objective, consisting of the reconstruction loss and the codebook commitment loss:

$$\mathcal{L}_{topo} = \mathbb{E}_{x \sim \mathcal{D}} \Big[ \|x - D_s(z_s)\|_2^2$$
$$+ \|\text{sg}[E_s(x)] - e\|_2^2 + \beta \|E_s(x) - \text{sg}[e]\|_2^2 \Big]. \quad (9)$$

This objective anchors the learned tangent bundle onto a valid data distribution, providing a stable geometric foundation for the subsequent dynamic modeling. The second term enforces the predicted tangent velocity $v_{amb}$ to adhere to the governing physical laws of the system. We formulate this generally as minimizing the norm of a *Physics Constraint Functional* $\mathcal{H}_{domain}(\cdot)$:

$$\mathcal{L}_{phy} = \mathbb{E}_{x \sim \mathcal{D}} \left[ \|\mathcal{H}_{domain}(x, v_{amb})\|_2^2 \right]. \quad (10)$$

Here, $\mathcal{H}_{domain}$ is an operator that measures the residual between the predicted geometric flow and the underlying physical truths. The specific form of this operator depends on the nature of the downstream task (shown in Table 1). For temporal evolution tasks such as ODE discovery or video analysis, $\mathcal{H}$ measures the integration or prediction error, evaluating the discrepancy between the time-integrated trajectory of $v_{amb}$ and the observed future states. Conversely, for systems with implicit dynamics like single-cell genomics, $\mathcal{H}$ measures the physical projection residual, evaluating the orthogonality of $v_{amb}$ with respect to the valid solution space defined by the governing reaction equations. This unified formulation ensures that regardless of the domain, the geometric flow is strictly grounded in its respective physical reality.

| Case | Prior | Ours |
|------|-------|------|
| **ODE** | $\min \sum_k \|x_{k+1} - \Phi_{\Delta t_k}(x_k; f_\Xi)\|_2^2$ | $\min \sum_k \|x_{k+1} - D_s(\Phi_{\Delta t_k}(z_k; \mathcal{V}))\|_2^2$ |
| **RNA** | $\min \sum_i \sum_{j \in \mathcal{N}(i)} w_{ij} \|\hat{v}_\theta(x_i) - \hat{v}_\theta(x_j)\|_2^2$ | $\min \sum_i \|M_i \theta_i^\star - \hat{v}_{\tan}(x_i)\|_2^2$ |
| **Video** | $\|I_{t+1} - \mathcal{W}(I_t, u_{\text{flow}}(I_t, I_{t+1}))\|_2^2$ | $\|\Delta I_{obs} - \text{JVP}(G, E_s(x_t), s_\theta(x_t))\|_2^2$ |

*Table 1.* **Domain Objectives across Three Cases.** For ODE, $\Phi_{\Delta t_k}(\cdot)$ denotes a one-step integrator with step $\Delta t_k$; $f_\Xi$ is the velocity function with library, $\mathcal{V}$ is the latent velocity function $\mathcal{V}$ and $D_s$ is the state decoder. For RNA, $\hat{v}_\theta$ is the RNA velocity function with parameter $\theta$, $\hat{v}_{\tan}(x_i)$ is the tangent-projected velocity, $M$ is the kinetic design matrix, and $\theta_i^\star$ is the ridge solution used to form the projection residual. For Video, $u_{\text{flow}}$ is optical flow, $\mathcal{W}$ denotes warping, $\Delta I_{\text{obs}}$ is the observed pixel displacement, and JVP is the Jacobian-vector product of the generator $G$ along the score direction $s_\theta$ produced by a pre-trained diffusion model.

| Method | MSE ↓ | VPT ↑ | Discovered ODE |
|--------|-------|-------|----------------|
| **Ground Truth** | - | - | $\dot{x}_1 = -1.0x_2, \; \dot{x}_2 = 1.0x_1$ |
| SINDy | 1.2902 | 3.8 | $\dot{x}_1 = -0.5993x_1, \; \dot{x}_2 = -0.6179x_2$ |
| IRK-SINDy | 1.2926 | 3.8 | $\dot{x}_1 = 0.7119x_2, \; \dot{x}_2 = -0.8269x_1$ |
| MNN | 4.8494 | 3.8 | $\dot{x}_1 = 0.5156x_2, \; \dot{x}_2 = 0$ |
| S-MNN | 38.3503 | 3.8 | $\dot{x}_1 = -0.7023x_1, \; \dot{x}_2 = 0.6061x_2$ |
| **GFG (Ours)** | **0.0003** | **26.1** | $\dot{x}_1 = -0.9737x_2, \; \dot{x}_2 = 1.0188x_1$ |

*Table 2.* **The Discovery of Ordinary Differential Equation.** The observed data are sampled from 2D circle rotation physics system under large, non-uniform $\Delta t \in [2.106, 4.161]$. We report one-step rollout MSE and valid prediction time (VPT). The ground truth equation is shown in the gray row for reference.

## 4. Experiments

We evaluate Geometric Flow Grounding across three distinct regimes, ranging from theoretical physical systems to high-dimensional biological data and generative media.

### 4.1. Case 1: Ordinary Differential Equation Discovery

Equation discovery aims to infer closed-form governing laws from observed data. In this paper, we follow the typical Sparse Identification of Nonlinear Dynamics (SINDy) setting(Brunton et al., 2016), which achieves the discovery through the sparse regression over a predefined library of candidate functions.

**Setup.** We simulate *uniform circular motion* under non-uniform, sparse temporal sampling (Figure 3a) to benchmark GFG against classical SINDy (Brunton et al., 2016) and IRK-SINDy (Anvari et al., 2025), as well as two recent neural-network-based methods, MNN (Pervez et al., 2024) and S-MNN (Chen et al., 2025a), built on a differentiable linear-programming solver. Under this challenge sampling scenario, classical SINDy fails due to derivative estimation errors. IRK-SINDy tries to avoid this by recovering equations through minimizing simulation loss via implicit Runge-Kutta schemes. However, it remains prone to instability and ambiguity with large gaps.

| Method | MouseBrain | | | Dentate Gyrus | | | Retina | | | Erythroid Lineage | | |
|---|---|---|---|---|---|---|---|---|---|---|---|---|
| | VeloCoh ↑ | ICCoh ↑ | CBdir ↑ | VeloCoh ↑ | ICCoh ↑ | CBdir ↑ | VeloCoh ↑ | ICCoh ↑ | CBdir ↑ | VeloCoh ↑ | ICCoh ↑ | CBdir ↑ |
| scVelo (Bergen et al., 2020) | 0.765 ±0.000 | 0.504 ±0.002 | 0.435 ±0.011 | 0.823 ±0.000 | 0.673 ±0.000 | 0.254 ±0.018 | 0.922 ±0.013 | 0.743 ±0.015 | 0.333 ±0.027 | 0.627 ±0.000 | 0.495 ±0.016 | 0.067 ±0.001 |
| veloVI (Gayoso et al., 2024) | 0.620 ±0.000 | 0.318 ±0.000 | 0.269 ±0.034 | 0.649 ±0.000 | 0.436 ±0.000 | **0.621** ±0.043 | 0.651 ±0.000 | 0.371 ±0.000 | 0.185 ±0.062 | 0.484 ±0.000 | 0.320 ±0.000 | 0.173 ±0.023 |
| DeepVelo (Cui et al., 2024) | 0.588 ±0.012 | 0.364 ±0.013 | 0.411 ±0.146 | 0.868 ±0.018 | 0.783 ±0.022 | 0.285 ±0.347 | 0.680 ±0.008 | 0.481 ±0.014 | 0.377 ±0.085 | 0.702 ±0.016 | 0.569 ±0.017 | 0.216 ±0.128 |
| TIvelo (Ge et al., 2025) | 0.385 ±0.000 | 0.133 ±0.000 | 0.515 ±0.012 | 0.746 ±0.000 | 0.422 ±0.001 | 0.413 ±0.000 | 0.675 ±0.030 | 0.244 ±0.018 | 0.384 ±0.048 | 0.329 ±0.000 | 0.157 ±0.000 | 0.539 ±0.009 |
| Graphvelo (Chen et al., 2025b) | 0.683 ±0.000 | 0.366 ±0.000 | -0.143 ±0.031 | 0.790 ±0.000 | 0.696 ±0.000 | 0.427 ±0.148 | 0.759 ±0.015 | 0.436 ±0.070 | 0.316 ±0.063 | 0.887 ±0.003 | 0.859 ±0.005 | 0.569 ±0.003 |
| **GFG (Ours)** | **0.864** ±0.036 | **0.736** ±0.078 | **0.657** ±0.055 | **0.946** ±0.014 | **0.908** ±0.018 | 0.473 ±0.030 | **0.998** ±0.001 | **0.997** ±0.001 | **0.430** ±0.025 | **0.994** ±0.000 | **0.989** ±0.000 | **0.746** ±0.003 |

*Table 3.* **Quantitative comparison across four scRNA-seq datasets.** We report three key metrics: VeloCoh for global smoothness, ICCoh for local consistency, and CBDir for biological accuracy. Our GFG consistently outperforms state-of-the-art baselines, especially in geometric consistency metrics. **Bold** and underline denote the best and second-best results, respectively. Results are reported over 5 runs.

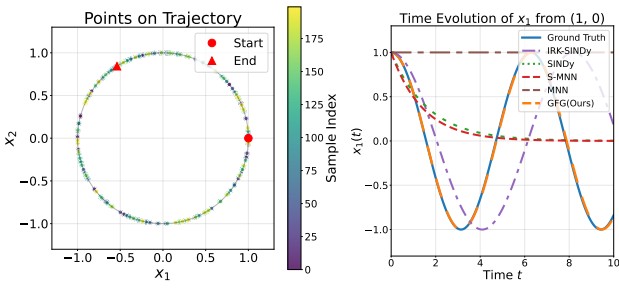

*(a) Dataset Visualization*     *(b) Trajectory Comparison*

*Figure 3.* **Dynamics Discovery on the Circular Motion System.** (a) Visualization of sampled states from a uniform circular-motion system under sparse, non-uniform temporal sampling. Point color encodes the sample index (purple → yellow indicates earlier → later samples). The red circle and triangle mark the start and end states, respectively. (b) Rollout of $x_1(t)$ from the start state $(1, 0)$. GFG closely matches the ground truth, whereas SINDy and S-MNN converge toward zero, MNN remains near the initial state, and IRK-SINDy exhibits phase drift.

**Metrics.** We evaluate performance from three aspects: 1) *Equation Recovery*, assessing whether the correct terms and coefficients are identified; 2) *One-Step Rollout MSE*, measuring local predictive accuracy; and 3) *Valid Prediction Time* (VPT), quantifying long-term stability before the trajectory diverges. Details on metric calculations are provided in Appendix B.1.2.

**Comparison with the State-of-The-Art Baselines.** Table 2 summarizes the quantitative results. GFG significantly outperforms both baselines, achieving much lower MSE and longer VPT. Crucially, GFG is the only method that correctly recovers the exact equation coefficients. As shown in Figure 3(b), classical SINDy fails to maintain the circular motion, learning incorrect terms that cause the trajectory to decay rapidly toward the center. IRK-SINDy, while keeping the circular shape, learns coefficients with reversed signs. This error comes from the *phase ambiguity* in sparse sam-

pling: when the time gap is large (rotation $> \pi$), the data can be explained by multiple different paths. Without geometric constraints, integral-based methods often pick a plausible but wrong solution. The neural baselines exhibit the same difficulty. MNN recovers an incomplete system where the $\dot{x}_2$ component collapses; S-MNN learns decoupled self-terms with a sign flip that turns the conservative rotation into a divergent system, yielding the largest error in Table 2. GFG uses manifold constraints to resolve this ambiguity, ensuring the trajectory stays aligned with the ground truth. More complicated cases are shown in Appendix D.1

### 4.2. Case 2: Single-Cell RNA Velocity Inference

Single-cell RNA velocity inference aims to infer continuous dynamic trajectories from static gene expression snapshots (La Manno et al., 2018), serving as a critical tool for estimating cell state transition probabilities. However, the high dimensionality, sparsity, and noise inherent in single-cell data pose significant challenges to this task.

**Baselines.** We compare GFG with five state-of-the-art methods, including the stochastic model scVelo (Bergen et al., 2020), the generative framework veloVI (Gayoso et al., 2024), and recent graph-based approaches like DeepVelo (Cui et al., 2024), TIvelo (Ge et al., 2025), and Graphvelo (Chen et al., 2025b). These methods typically rely on local smoothness heuristics in the ambient expression space, often yielding velocity vectors misaligned with the cellular manifold. While Graphvelo attempts a post-hoc geometric correction, GFG explicitly projects the learned flow onto the tangent bundle via end-to-end training, ensuring geometric validity in an intrinsic way.

**Datasets and Metrics.** We evaluate GFG on four standard scRNA-seq benchmarks representing diverse developmental topologies: *MouseBrain* (Ge et al., 2025), *Dentate*

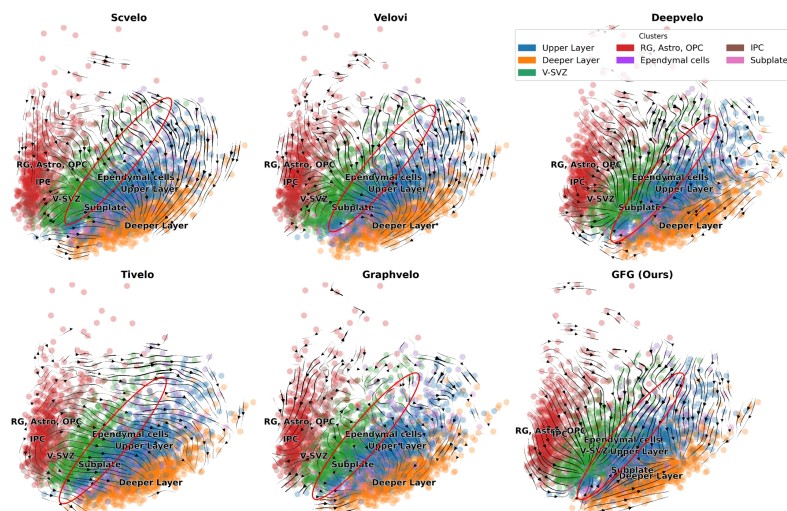

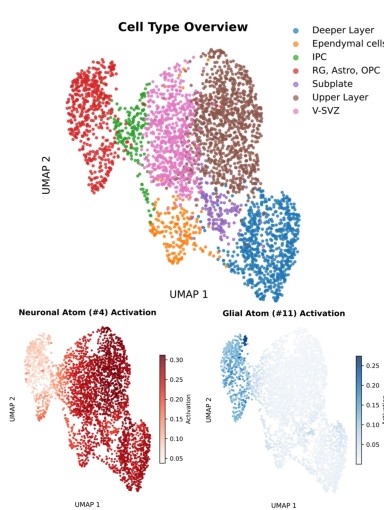

Figure 4. **RNA velocity on Mouse Brain.** Red ellipses highlight spurious V-SVZ (green) to Upper Layer (blue) transitions in baselines. GFG suppresses these hallucinations by enforcing manifold geometry, ensuring trajectories respect biological boundaries.

Figure 5. **Spatial disentanglement on Mouse Brain.** This visualizes GFG's autonomous separation of competing cell fates into orthogonal flows.

*Gyrus* (Hochgerner et al., 2018), *Retina* (Lo Giudice et al., 2019), and *Erythroid Lineage* (Pijuan-Sala et al., 2019). We evaluate performance using three complementary geometric and biological metrics: 1) *VeloCoh* (Velocity Coherence), measuring the global smoothness of the inferred vector field; 2) *ICCoh* (Intraclass Correlation), assessing the local directional agreement among neighboring cells; and 3) *CBDir* (Cross-Boundary Directionality), verifying the correctness of transitions along biological lineages. Details in Appendix.

**Comparison with the State-of-The-Art Baselines.** Table 3 shows that GFG dominates geometric metrics, particularly achieving significant Recall gains. These quantitative improvements are qualitatively supported by Figure 4. In the red-highlighted regions of the Mouse Brain dataset, baseline methods produce spurious transitions from the V-SVZ to the Upper Layer due to their reliance on ambient space smoothness. By contrast, GFG enforces tangent bundle projection, effectively suppressing these biologically impossible shortcuts by ensuring inferred trajectories respect known cellular boundaries and manifold curvature. This confirms that GFG captures intrinsic dynamical laws that remain elusive to conventional statistical or smoothing approaches.

**Dynamics Discovery and Interpretability.** We examine the biological semantics of GFG's velocity codebook on the Mouse Brain dataset. By decoding discrete atoms for GO enrichment, we directly verify their lineage specificity. Examples are shown in Figure 5, where the learned atoms demonstrate clear spatial disentanglement: *Atom 4* tracks the neuronal trajectory (V-SVZ to cortex), whereas *Atom 11* specifically highlights the glial branch (RG, Astro, OPC). This geometric separation is substantiated by statistical ev-

| Method | Recall ↑ | Accuracy ↑ | F1-Score ↑ | AUROC ↑ |
|---|---|---|---|---|
| STIL (Gu et al., 2021) | 27.0 | 63.5 | 35.8 | 93.5 |
| NPR (Tan et al., 2024) | 57.4 | 78.0 | 68.4 | 93.0 |
| TALL (Xu et al., 2023) | 60.8 | 79.9 | 72.6 | 95.7 |
| DeMamba (Chen et al., 2024) | 72.0 | 84.2 | 80.1 | 93.9 |
| NSG-VD (Zhang et al., 2025) | 88.0 | 91.5 | **90.9** | 96.1 |
| **GFG (Ours)** | 73.6 | 88.2 | 81.5 | 93.4 |
| **GFG + NSG-VD** | **93.2** | **92.2** | 88.9 | **96.3** |

Table 4. **Quantitative comparison on AI-Generated Video Detection (Average).** We report the average performance across ten diverse video generation benchmarks.

idence (Appendix D.2.1). The neuronal atom is enriched in *nervous system development* ($p = 7.30 \times 10^{-6}$), reflecting neurogenesis. In contrast, the glial atom aligns with structural functions like *extracellular matrix organization* ($p = 2.85 \times 10^{-4}$), capturing the distinct scaffolding role of glial cells. This confirms that GFG captures the underlying modularity of developmental systems, bridging geometric inference with biological insight.

### 4.3. Case 3: AI-Generated Video Detection

The detection of AI-generated videos aims to distinguish real-world videos videos from those synthesized by text-to-video models. A core challenge in this domain is generalization, as detectors trained on specific generators often fail when encountering unseen synthesis algorithms. We frame this task as a geometric consistency problem where we find that real videos evolve along a continuous manifold while generated content exhibits detectable geometric irregularities regardless of the specific generator architecture.

**Geometric Consistency Analysis.** We analyze the distribution of normalized tangent projection errors to investigate the geometric validity of video trajectories. To

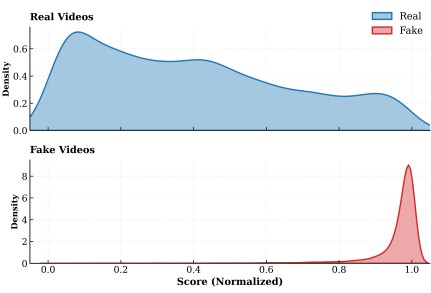

*Figure 6.* **Geometric consistency analysis.** We compare the density of normalized tangent projection errors for real (top) and generated (bottom) videos. While real videos generally align with the predicted manifold flow, generated content exhibits a sharp peak at the maximum deviation.

| Dataset | Variant | VeloCoh ↑ | ICCoh ↑ | CBDir ↑ |
|---|---|---|---|---|
| | w/o JVP | $0.993\pm0.032$ | $0.970\pm0.021$ | $0.405\pm0.048$ |
| *Pancreas* | w/o VQ | $0.993\pm0.027$ | $0.972\pm0.018$ | $0.125\pm0.051$ |
| | **GFG (Full)** | $\mathbf{0.994}\pm0.025$ | $\mathbf{0.976}\pm0.009$ | $\mathbf{0.588}\pm0.047$ |
| | w/o JVP | $0.695\pm0.299$ | $0.594\pm0.303$ | $0.465\pm0.282$ |
| *Mouse Brain* | w/o VQ | $0.587\pm0.361$ | $0.457\pm0.365$ | $0.106\pm0.482$ |
| | **GFG (Full)** | $\mathbf{0.864}\pm0.036$ | $\mathbf{0.736}\pm0.078$ | $\mathbf{0.657}\pm0.055$ |

*Table 5.* **Ablation study on Pancreas and Mouse Brain.** JVP and VQ contribute complementary benefits: JVP improves the geometric validity of the learned vector field, while VQ enhances mode disentanglement and robustness to noisy local trends.

quantify this, we measure the discrepancy between the observed pixel displacement $\Delta I_{obs} = I_{t+1} - I_t$ and the theoretically predicted manifold flow. The latter is derived by projecting the estimated latent score vector $v_{score}$ back to pixel space via the decoder's Jacobian: $\Delta I_{pred} = J_{Decoder}(z_t) \cdot v_{score}$. The final detection score is the normalized distance $||\Delta I_{obs} - \Delta I_{pred}||^2$. As shown in Figure 6, generated videos exhibit a distinct peak at maximum deviation (1.0), confirming that they violate intrinsic geometric constraints despite high static fidelity.

**Datasets and Metrics.** Following NSG-VD (Zhang et al., 2025), we utilize a balanced training set from Gen-Video (Chen et al., 2024), comprising 10,000 real videos from Kinetics-400 (Kay et al., 2017) and 10,000 generated samples from Pika (Pika Art, 2022). To evaluate generalization to unseen synthesis algorithms, our validation and test sets strictly exclude the training generator. We report performance using four standard metrics: Recall, Accuracy, F1-Score, and AUROC.

**Comparison with the State-of-The-Art Baselines.** Table 4 shows that GFG provides a complementary geometric signal for AI-generated video detection. Rather than replacing NSG-VD, which monitors spatiotemporal gradient distribution shifts via Maximum Mean Discrepancy, GFG verifies whether temporal evolution is locally aligned with the tangent flow induced by the learned generative manifold. This difference is important for high-fidelity generators, where fake videos may match global real-video statistics but

| (a) Sensitivity to Codebook Size $L$ | | | (b) Sensitivity to $\lambda$ | | | |
|---|---|---|---|---|---|---|
| Size $L$ | VeloCoh ↑ | ICCoh ↑ | CBDir ↑ | Weight $\lambda$ | VeloCoh ↑ | ICCoh ↑ | CBDir ↑ |
| 8 | 0.996 | 0.987 | 0.579 | 0 | 0.993 | 0.970 | 0.405 |
| 16 | 0.994 | 0.980 | 0.580 | 1 | 0.993 | 0.976 | 0.588 |
| 32 | 0.994 | 0.976 | 0.588 | 10 | 0.993 | 0.976 | 0.588 |
| 64 | 0.993 | 0.976 | 0.580 | 50 | 0.992 | 0.972 | 0.579 |

*Table 6.* **Sensitivity Analysis on the Pancreas dataset.** (a) GFG shows high robustness to codebook size $L$, with biological accuracy (CBDir) peaking at $L = 32$. (b) Incorporating the geometric weight $\lambda > 0$ significantly boosts performance compared to the baseline ($\lambda = 0$), maintaining stability across a wide range.

still exhibit off-manifold motion patterns that violate plausible physical continuity. As a result, combining GFG with NSG-VD achieves the best overall performance, improving Recall by 5.2% over NSG-VD and reaching an AUROC of 96.3%. Although the margin over NSG-VD is moderate and NSG-VD retains a slightly higher F1-score, these results suggest that tangent-space consistency is a useful threshold-independent criterion and is most effective when used as a complementary verification signal.

### 4.4. Ablation and Sensitivity

Table 5 evaluates the contributions of the two main components of GFG on both Pancreas and Mouse Brain. The results show that JVP and VQ play complementary roles. Removing the JVP-based geometric constraint consistently decreases CBDir, indicating that tangent-space alignment helps preserve geometrically valid and biologically meaningful trajectories on the learned manifold. Removing the VQ codebook causes a larger degradation in CBDir, from $0.588$ to $0.125$ on Pancreas and from $0.657$ to $0.106$ on Mouse Brain, suggesting that VQ is particularly important for disentangling heterogeneous transition modes and improving robustness to noisy local trends. Combining both components yields the strongest performance across datasets.

**Hyperparametric Sensitivity**. Table 6 evaluates GFG's robustness to codebook size $L$ and geometric weight $\lambda$. Performance remains stable across various scales of $L$, with biological accuracy peaking at $L = 32$. Notably, incorporating the geometric constraint ($\lambda > 0$) yields a significant performance leap over the baseline ($\lambda = 0$), while maintaining consistent results across $\lambda \in [1, 10]$. This confirms that GFG is robust and insensitive to hyperparameter tuning within a practical range.

## 5. Related Work

**Learning Continuous Dynamics from Snapshots.** Learning continuous dynamics from snapshot observations intersects with several continuous-time modeling paradigms. Neural ODEs (Chen et al., 2018) provide a foundation by parameterizing vector fields to infer dynamics from irregular measurements, a framework extended to partially observed and multi-agent systems to resolve hidden state trajectories.

A parallel line of work pursues explicit equation discovery: MNN (Pervez et al., 2024) and its scalable variant S-MNN (Chen et al., 2025a) embed differentiable mechanistic solvers into neural networks to recover governing ODEs, with MechNN-PDE (Pervez et al., 2025) further extending this paradigm to PDEs. Complementary approaches focus on reconstructing density evolution directly from snapshot distributions. For instance, Action Matching (Neklyudov et al., 2023) and optimal-transport-based flows (Kawano et al., 2025) estimate continuous trajectories from temporal marginals or sparse snapshots without full path information. Together, these methods address the challenges of identifiability and temporal sparsity, providing a theoretical basis for our approach to recover latent manifold flows from discrete snapshot data.

**RNA Velocity Inference.** RNA velocity modeling has evolved from solving transcriptional dynamics via scVelo (Bergen et al., 2020) to deep learning frameworks like veloVI (Gayoso et al., 2024) and DeepVelo (Cui et al., 2024) that resolve complex kinetics. To enhance stability, recent approaches like TIvelo (Ge et al., 2025) and Graphvelo (Chen et al., 2025b) introduce trajectory guidance or graph-based tangent projections. Unlike these, GFG ensures consistency by strictly projecting velocity vectors onto the manifold's intrinsic tangent bundle via Jacobian-Vector Products.

**Video Forgery Detection.** Video detection has transitioned from supervised learning models such as DeMamba (Chen et al., 2024), STIL (Gu et al., 2021), and TALL (Xu et al., 2023), to artifact analysis in frameworks like NPR (Tan et al., 2024) and D3 (Zheng et al., 2025). More recently, physics-driven paradigms like NSG-VD (Zhang et al., 2025) and motion-based CNNs (Amerini et al., 2019) detect violations of physical conservation or continuity. In contrast, GFG provides a deterministic geometric criterion by verifying the alignment between observed trajectories and the manifold's theoretical tangent flow.

## 6. Conclusion and Future Work

In this paper, we introduced Geometric Flow Grounding (GFG), a unified framework that resolves the critical challenge of off-manifold hallucinations in continuous dynamics learning. We achieve this via two synergistic mechanisms: the Neural Tangent Projection, which eliminates integration error by strictly confining evolution to the tangent bundle via matrix-free Jacobian-Vector Products, and a Dynamics Primitive Codebook, which mitigates smoothing drift by quantizing conflicting flows into interpretable, distinct geometric modes. Empirical evaluations across physics, biology, and computer vision validate that this framework serves as both a numerical stabilizer for identifying governing equations and a mechanism for disentangled discovery.

**Future Work.** Geometric Flow Grounding could be further developed into a transferable inference-time module for generative foundation models. For example, it may serve as a *plug-and-play operator* for video diffusion models, guiding generation trajectories toward valid tangent bundles to improve physical consistency without requiring full model retraining.

**Limitation.** Our geometric construction assumes that the learned generator is locally differentiable and well-conditioned, and that the dominant dynamics are approximately tangential to the learned manifold. Therefore, GFG is better suited to systems with continuous dominant transitions, but may be less appropriate for processes involving abrupt jumps, strong off-manifold effects, or dynamics that are poorly captured by a tangent-space prior. In such cases, alternative geometric constraints may be needed. In addition, the latent dynamics and the learned generator are not necessarily uniquely identifiable, since different latent parameterizations may induce similar vector fields in the observation space. Our goal is thus not to recover a unique latent decomposition, but to learn geometrically consistent and meaningful dynamics in the observed space.

## Acknowledgements

This work was supported by Open Research Fund of the State Key Laboratory of Multimodal Artificial Intelligence Systems (No.MAIS2025064), National Science and Technology Major Project (No.2022ZD0115101), the Hangzhou Postdoctoral Daily Funding Program (No.103140026582502, 2025), National Natural Science Foundation of China Project (No.624B2115 No.U21A20427), Project (No.WU2022A009) from the Center of Synthetic Biology and Integrated Bioengineering of Westlake University, and InnoHK Program.

## Impact Statement

This work contributes to geometric deep learning for modeling continuous dynamics in natural systems. In computational biology and physics, the proposed manifold-grounded framework may improve the reliability of system identification by discouraging biologically or physically implausible trajectories. More broadly, such geometric constraints could serve as scalable consistency checks for AI-generated videos, supporting detection of physically inconsistent synthetic content and offering a path toward incorporating physical validity into future generative models. These capabilities may help improve the trustworthiness of scientific discovery and generative AI systems, although their effectiveness depends on the validity of the underlying manifold and dynamical assumptions.

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

# A. Theoretical Proofs

In this section, we provide detailed derivations for the theoretical results presented in the main paper. We first prove Theorem 3.2, which characterizes the role of tangent projection in inference and training. We then discuss robustness under imperfect learned manifolds, followed by the geometric consistency of the Jacobian-vector product used in Neural Tangent Projection.

## A.1. Proof of Theorem 3.2

For simplicity, we omit the dependence on the latent state $z$ and write $J_\theta$ for the Jacobian of the learned generator $G_\theta$ at the encoded point. Let $\mathbf{P}_{J_\theta}$ denote the orthogonal projection onto $\mathrm{Range}(J_\theta)$, and let $I - \mathbf{P}_{J_\theta}$ denote the projection onto its orthogonal complement.

**Inference denoising.** For any ambient estimator $\hat{f}_{\mathrm{amb}}$ and ground-truth vector field $f^*$, decompose $f^*$ into tangent and normal components with respect to the learned tangent space:

$$f^* = \mathbf{P}_{J_\theta} f^* + (I - \mathbf{P}_{J_\theta})f^*. \tag{11}$$

Then,

$$\mathbf{P}_{J_\theta} \hat{f}_{\mathrm{amb}} - f^* = \mathbf{P}_{J_\theta} \hat{f}_{\mathrm{amb}} - \mathbf{P}_{J_\theta} f^* - (I - \mathbf{P}_{J_\theta})f^* \tag{12}$$

$$= \mathbf{P}_{J_\theta}(\hat{f}_{\mathrm{amb}} - f^*) - (I - \mathbf{P}_{J_\theta})f^*. \tag{13}$$

The first term lies in $\mathrm{Range}(J_\theta)$, while the second term lies in its orthogonal complement. Hence, by the Pythagorean theorem,

$$\|\mathbf{P}_{J_\theta} \hat{f}_{\mathrm{amb}} - f^*\|_2^2 = \|\mathbf{P}_{J_\theta}(\hat{f}_{\mathrm{amb}} - f^*)\|_2^2 + \|(I - \mathbf{P}_{J_\theta})f^*\|_2^2. \tag{14}$$

This proves Eq. (3) in the main text. The first term measures the remaining tangent-space estimation error, while the second term measures the component of the true dynamics not captured by the learned tangent space.

In the special case where $f^* \in \mathrm{Range}(J_\theta)$, the normal residual vanishes:

$$(I - \mathbf{P}_{J_\theta})f^* = 0. \tag{15}$$

Therefore,

$$\|\mathbf{P}_{J_\theta} \hat{f}_{\mathrm{amb}} - f^*\|_2^2 = \|\mathbf{P}_{J_\theta}(\hat{f}_{\mathrm{amb}} - f^*)\|_2^2 \leq \|\hat{f}_{\mathrm{amb}} - f^*\|_2^2, \tag{16}$$

where the inequality follows from the non-expansiveness of orthogonal projection. Taking square roots gives

$$\|\mathbf{P}_{J_\theta} \hat{f}_{\mathrm{amb}} - f^*\|_2 \leq \|\hat{f}_{\mathrm{amb}} - f^*\|_2, \tag{17}$$

which proves Eq. (4).

**Manifold alignment.** For an observed transition $\Delta x$, the best tangent-space explanation is obtained by solving

$$\min_u \|\Delta x - J_\theta u\|_2^2. \tag{18}$$

This is a linear least-squares problem. Its solution projects $\Delta x$ onto the column space of $J_\theta$, i.e., onto $\mathrm{Range}(J_\theta)$. Therefore, the minimum residual is the squared norm of the component of $\Delta x$ orthogonal to this learned tangent space:

$$\min_u \|\Delta x - J_\theta u\|_2^2 = \|\Delta x - \mathbf{P}_{J_\theta} \Delta x\|_2^2 = \|(I - \mathbf{P}_{J_\theta})\Delta x\|_2^2. \tag{19}$$

This proves Eq. (5). In GFG, the physical consistency loss penalizes this projection residual, thereby encouraging the learned decoder Jacobian to span directions that explain observed transitions.

## A.2. Robustness under Imperfect Learned Manifolds

The guarantee in Theorem 3.2 is stated with respect to the learned tangent space induced by the decoder $G_\theta$. When the learned manifold is imperfect, the residual error depends on both the intrinsic non-tangential component of the true dynamics and the approximation error of the learned tangent space.

Let $\mathbf{P}_\star$ denote the orthogonal projection onto the tangent space of the underlying data manifold. Starting from the normal residual term in Eq. (3), we have

$$\|(I - \mathbf{P}_{J_\theta})f^*\|_2 = \|f^* - \mathbf{P}_{J_\theta}f^*\|_2 \tag{20}$$

$$\leq \|f^* - \mathbf{P}_\star f^*\|_2 + \|\mathbf{P}_\star f^* - \mathbf{P}_{J_\theta}f^*\|_2 \tag{21}$$

$$\leq \|(I - \mathbf{P}_\star)f^*\|_2 + \|\mathbf{P}_{J_\theta} - \mathbf{P}_\star\|_{\mathrm{op}}\|f^*\|_2. \tag{22}$$

The first term measures the intrinsic normal component of the true dynamics with respect to the underlying data manifold. The second term measures the discrepancy between the learned tangent space and the underlying tangent space.

We further relate this discrepancy to the local Jacobian approximation error. Let $J_\star$ denote the Jacobian of an underlying generator for the data manifold, and assume that $J_\star$ has rank $d$ with smallest nonzero singular value $\sigma_d(J_\star) > 0$. If the learned Jacobian satisfies

$$\|J_\theta - J_\star\|_{\mathrm{op}} \leq \delta_J, \tag{23}$$

and $\delta_J$ is sufficiently small relative to the local singular-value gap, then standard subspace perturbation bounds imply

$$\|\mathbf{P}_{J_\theta} - \mathbf{P}_\star\|_{\mathrm{op}} \leq C\frac{\delta_J}{\sigma_d(J_\star)}, \tag{24}$$

where $C$ is a constant depending on the local conditioning of the tangent space. Substituting this into the previous inequality gives

$$\|(I - \mathbf{P}_{J_\theta})f^*\|_2 \leq \|(I - \mathbf{P}_\star)f^*\|_2 + C\frac{\delta_J}{\sigma_d(J_\star)}\|f^*\|_2. \tag{25}$$

Consequently, the projected estimator error is controlled by three factors: the tangent-space component of the velocity estimation error, the intrinsic normal component of the true dynamics, and the local Jacobian approximation error of the learned generator. This formalizes that GFG does not require exact manifold recovery. Instead, its error degrades with the quality of the learned tangent space and the degree to which the true dynamics deviate from tangential evolution.

This also clarifies the role of the training objective. The topological reconstruction loss encourages the decoder manifold to approximate the observed data support, while the physical consistency loss penalizes transition components that cannot be explained by the learned tangent space. Thus, the learned tangent geometry is optimized jointly with the dynamics, rather than assumed to be exact or fixed in advance.

## A.3. Geometric Consistency of Neural Tangent Projection

**Proposition A.1** (*Geometric Consistency of JVP*). *Given a differentiable generator $G : \mathcal{Z} \to \mathcal{X}$ defining a learned manifold $\mathcal{M} = \mathrm{Image}(G)$, for any latent velocity $v_{\mathrm{lat}} \in \mathbb{R}^d$, the vector computed by the Jacobian-vector product*

$$v_{\mathrm{tan}} = \mathrm{jvp}(G, z, v_{\mathrm{lat}}) \tag{26}$$

*satisfies $v_{\mathrm{tan}} \in \mathcal{T}_{G(z)}\mathcal{M}$.*

*Proof.* Let $J_G(z) \in \mathbb{R}^{D \times d}$ be the Jacobian matrix of the generator $G$ at latent state $z$. By the definition of the learned manifold $\mathcal{M} = \mathrm{Image}(G)$, the tangent space at $x = G(z)$ is the column space of $J_G(z)$:

$$\mathcal{T}_{G(z)}\mathcal{M} = \{J_G(z)u \mid u \in \mathbb{R}^d\}. \tag{27}$$

In forward-mode automatic differentiation, the JVP operator computes the directional derivative of $G$ at $z$ along $v_{\mathrm{lat}}$:

$$\mathrm{jvp}(G, z, v_{\mathrm{lat}}) \triangleq \frac{d}{d\epsilon}G(z + \epsilon v_{\mathrm{lat}})\Big|_{\epsilon=0} = J_G(z)v_{\mathrm{lat}}. \tag{28}$$

Since $v_{\text{lat}} \in \mathbb{R}^d$, the resulting vector $J_G(z)v_{\text{lat}}$ is a linear combination of the columns of $J_G(z)$. Therefore,

$$v_{\text{tan}} \in \text{Range}(J_G(z)) = \mathcal{T}_{G(z)}\mathcal{M}. \tag{29}$$

Thus, the JVP construction ensures that the instantaneous velocity lies in the tangent space of the learned manifold. $\square$

## B. Datasets and Evaluation Metrics

### B.1. Case I: ODE Discovery

#### B.1.1. DATASET PREPARATION

We construct a synthetic ODE-discovery benchmark from *uniform circular motion* on the unit circle. The ground-truth dynamics are $\dot{x}_1 = -\omega x_2, \dot{x}_2 = \omega x_1$, with $\omega = 1$ and initial state $x_0 = (1, 0)$. To ensure that the generated states lie exactly on the manifold $\mathbb{S}^1$, we parameterize the trajectory as $x_k = (\cos\theta_k, \sin\theta_k)$ and generate a sequence of $n = 200$ observations. We emulate sparse and irregular timestamps by sampling angular increments between consecutive observations. For $k = 0, \ldots, n - 2$, we draw i.i.d. $\Delta\theta_k \sim \text{Uniform}(\theta_{\min}, \theta_{\max}), (\theta_{\min}, \theta_{\max}) = (120°, 240°)$, and update $\theta_{k+1} = \theta_k + \Delta\theta_k$. The non-uniform time gaps are defined by $\Delta t_k = \Delta\theta_k/\omega$, and timestamps are accumulated as $t_{k+1} = t_k + \Delta t_k$ with $t_0 = 0$. This construction yields highly sparse transitions, where consecutive observations can be separated by large angular displacements (often exceeding $\pi$), making continuous-time identification from discrete snapshots particularly challenging. We split the resulting sequence into $60\%/40\%$ for training and testing, i.e., the first $n_{\text{train}} = 120$ samples are used for training or equation discovery and the remaining $n_{\text{test}} = 80$ samples are reserved for evaluation. For methods that require non-uniform step sizes, we provide $\Delta t_k$ alongside the state observations.

#### B.1.2. EVALUATION METRICS

Methods are trained on the first $60\%$ of the trajectory (the training split) and evaluated on the remaining $40\%$ (the test split). For all methods, a uniform second-order polynomial library $\{x_1, x_2, x_1^2, x_1 x_2, x_2^2\}$ is provided. In this experiment, models are provided with the observed states $\{x_k\}$ together with their associated non-uniform time gaps $\{\Delta t_k\}$. We evaluate all methods using two complementary metrics computed on the *test split*:

(1) *One-step rollout MSE* measures short-horizon predictive accuracy of the recovered dynamics: for each consecutive pair in the test segment, we roll out the learned ODE for a single step from $x_k$ over $\Delta t_k$ to obtain $\hat{x}_{k+1}$, and report

$$\text{MSE}_{\text{1-step}} = \frac{1}{N_{\text{test}} - 1} \sum_{k=0}^{N_{\text{test}}-2} \left\| \hat{x}_{k+1} - x_{k+1} \right\|_2^2. \tag{30}$$

(2)*Valid prediction time (VPT)* quantifies long-horizon stability: starting from the first test state, we iteratively roll out the recovered dynamics and compute the relative error

$$e(t_k) = \frac{\|\hat{x}(t_k) - x(t_k)\|_2}{\|x(t_k)\|_2}, \tag{31}$$

then define

$$\text{VPT}_\varepsilon = \min\{t_k : e(t_k) > \varepsilon\}, \tag{32}$$

where $\varepsilon$ is a fixed error threshold (set to $\varepsilon = 0.1$ for our experiments).

### B.2. Case II: Single-Cell RNA Velocity

#### B.2.1. DATASET DESCRIPTION

**Embryonic mouse brain** (Ge et al., 2025): The embryonic mouse brain data from 10x Genomics is accessible at 10x website. This dataset contains single-cell RNA sequencing data from 4,881 cells of the E18 mouse brain.

**Dentate gyrus development** (Hochgerner et al., 2018): The dentate gyrus development dataset from the Gene Expression Omnibus (GSE95753). Its subset is accessible via the `scvelo.datasets.dentategyrus` function in Python. This dataset contains single-cell RNA sequencing data from 2,930 cells of the mouse dentate gyrus.

**Mouse retina development** (Lo Giudice et al., 2019): The mouse retina development dataset (GSM3466902) is available from Kharchenko Lab . This dataset contains single-cell RNA sequencing data from 2,726 cells of the developing mouse retina.

**Mouse gastrulation (erythroid)** (Pijuan-Sala et al., 2019): The mouse gastrulation erythroid dataset is accessible via the `scvelo.datasets.gastrulation_erythroid` function in Python. This dataset contains single-cell RNA sequencing data from 9,815 cells of the mouse gastrulation erythroid lineage.

**Pancreatic endocrinogenesis** (Bastidas-Ponce et al., 2019): The pancreatic endocrinogenesis dataset (GSE132188) is accessible via the `scvelo.datasets.pancreas` function in Python. This dataset contains single-cell RNA sequencing data from 3,696 cells of the developing mouse pancreas.

### B.2.2. METRICS FOR EVALUATING

To evaluate the performance of RNA velocity models, we consider three main metrics: Velocity Coherence (VeloCoh), Inter-Cluster Coherence (ICCoh), and Cross-Boundary Directionality (CBdir). Each metric captures different aspects of the model's predictive quality.

**1. Velocity Coherence (VeloCoh)** VeloCoh measures the reliability of the predicted velocity vector for each cell, reflecting how consistent the velocity directions are within a cell's local neighborhood.

For each cell $i$, the velocity confidence is computed as the mean cosine similarity between the cell's velocity vector and the velocity vectors of its neighbors:

$$\text{velocity\_confidence}_i = \max\left(0, \frac{1}{|N(i)|} \sum_{j \in N(i)} \frac{\tilde{V}_i \cdot \tilde{V}_j}{\|\tilde{V}_i\| \, \|\tilde{V}_j\|}\right), \tag{33}$$

where

$$\tilde{V}_i = V_i - \frac{1}{G} \sum_{g=1}^{G} V_{ig}$$

is the centered velocity vector of cell $i$, $N(i)$ is the set of neighboring cells, and $\|\cdot\|$ denotes the Euclidean norm.

The overall VeloCoh score for the dataset can be defined as the average of all single-cell velocity confidences:

$$\text{VeloCoh} = \frac{1}{N} \sum_{i=1}^{N} \text{velocity\_confidence}_i, \tag{34}$$

where $N$ is the total number of cells in the dataset.

**2. Inter-cluster Coherence (ICCoh)** Inter-cluster Coherence (ICCoh) quantifies the alignment of velocity vectors within each cluster, reflecting how consistent the velocity directions are among cells of the same cluster.

Let $C_k$ denote the set of cells in cluster $k$, and let $\mathbf{v}_i$ be the velocity vector of cell $i$ in $C_k$. We first normalize the velocity vectors to unit length:

$$\hat{\mathbf{v}}_i = \frac{\mathbf{v}_i}{\|\mathbf{v}_i\|}, \quad \forall i \in C_k \tag{35}$$

The cluster-level coherence is computed as the mean cosine similarity between all pairs of cells within the cluster:

$$\text{coherence}(C_k) = \frac{2}{|C_k|(|C_k| - 1)} \sum_{i,j \in C_k, i<j} \hat{\mathbf{v}}_i \cdot \hat{\mathbf{v}}_j \tag{36}$$

Finally, the overall ICCoh score for the dataset is the weighted average of cluster coherences, weighted by cluster size:

$$\text{ICCoh} = \frac{\sum_k |C_k| \, \text{coherence}(C_k)}{\sum_k |C_k|} \tag{37}$$

Here, $|C_k|$ is the number of cells in cluster $k$, and the dot product $\hat{\mathbf{v}}_i \cdot \hat{\mathbf{v}}_j$ represents the cosine similarity between velocity vectors of cells $i$ and $j$ within the same cluster. ICCoh ranges from 0 to 1, with higher values indicating more coherent velocity directions within clusters.

**3. Cluster Boundary Directionality (CBDir)**  CBDir quantifies how well the velocity vectors of cells in a source cluster align with the direction toward neighboring target clusters along defined cluster edges.

Let $\mathcal{E}$ denote the set of directed cluster edges $(u, v)$, where $u$ is the source cluster and $v$ the target cluster. For each edge $(u, v) \in \mathcal{E}$, let $C_u$ and $C_v$ be the sets of cells in clusters $u$ and $v$, respectively. Denote $\mathbf{x}_i$ as the embedding coordinate (e.g., UMAP) of cell $i$ and $\mathbf{v}_i$ as its velocity vector. For each cell $i \in C_u$, define its boundary neighbors in $C_v$ as

$$B_i^{(v)} = \{j \in C_v \mid j \text{ is a nearest neighbor of } i\}.$$

The global CBDir score is then computed as a nested average over boundary neighbors, cells, and cluster edges:

$$\text{CBDir} = \frac{1}{|\mathcal{E}|} \sum_{(u,v) \in \mathcal{E}} \frac{1}{|C_u|} \sum_{i \in C_u} \frac{1}{|B_i^{(v)}|} \sum_{j \in B_i^{(v)}} \frac{\mathbf{v}_i \cdot (\mathbf{x}_j - \mathbf{x}_i)}{\|\mathbf{v}_i\| \, \|\mathbf{x}_j - \mathbf{x}_i\|}.$$

Higher CBDir values indicate that cells' velocity vectors are well-aligned toward neighboring clusters along cluster edges, reflecting coherent directional transitions between clusters.

### B.3. Case III: AI-Generated Video Detection

To ensure the robustness of the detector, we utilize a large-scale dataset compiled from various generative sources and real-world repositories, strictly following the dataset partitioning and testing protocol used by NSG-VD (Zhang et al., 2025) and related works:

- **Training Set:** We use a subset of the GenVideo  dataset. The training set includes 10000 real videos from Kinetics-400(val) (Kay et al., 2017) and 10000 fake videos generated by Pika (Pika Art, 2022).

- **Validation Set:** The validation set is partitioned to test generalization to unseen generators. It includes 100 real videos (MSR-VTT (Xu et al., 2016)) and 700 fake videos from generators not seen during training, such as MoonValley (Moonvalley AI, 2022) and MorphStudio (Morph Studio, 2023).

- **Test Set:** We follow a rigorous testing protocol using 500 real videos (MSR-VTT) and 500 fake videos per generator across multiple categories (*e.g.*, Sora (Brooks et al., 2024), Gen-2 (Runway Research, 2023), HotShot (Hotshot, 2023), Lavie (Wang et al., 2025)).

## C. Network Architectures and Training Details

In this section, we detail the mathematical formulation of the domain-specific guidance terms ($\mathcal{L}_{task}$), network architectures and training methods used in our experiments.

### C.1. General Architecture

**Generator / Decoder.** For the Lorenz system, we use a 4-layer MLP with Tanh activation. For scRNA-seq data, we use a symmetric 4-layer MLP with ReLU activation. For Video data (Case 3), we utilize the pre-trained Stable Video Diffusion (SVD) decoder, which is kept frozen during inference.

**Dual-Stream Encoders.** Both $E_s$ (State) and $E_v$ (Velocity) share a similar architecture backbone (ResNet-18 for video, 3-layer MLP for vectors) but diverge at the quantization heads.

- **Topological Codebook:** Size $K = 512$, dimension $d = 64$.

- **Motif Codebook:** Size $L = 16$ (as determined by sensitivity analysis), dimension $d = 64$.

## C.2. Case I: ODE Discovery

### C.2.1. IMPLEMENTATION

Many scientific processes can be described by nonlinear ordinary differential equations (ODEs), and a central goal in data-driven equation discovery is to identify these governing laws directly from measured trajectories. Among existing approaches, sparse identification of nonlinear dynamics (SINDy) is widely used due to its ability to produce interpretable closed-form vector fields via sparse regression over a predefined library of candidate functions. However, standard SINDy typically depends on accurate time-derivative estimates; when timestamps are sparse, finite-difference approximations become unreliable and can lead to incorrect model selection.

**Problem Formalization.** We consider a nonlinear dynamical system governed by an ordinary differential equation (ODE):

$$\dot{\mathbf{x}}(t) = \mathbf{f}(\mathbf{x}(t)), \qquad \mathbf{x}(t) \in \mathbb{R}^d, \tag{38}$$

where $\mathbf{f} : \mathbb{R}^d \to \mathbb{R}^d$ is an unknown vector field. We are given state sampled along trajectories at discrete timestamps $\{t_k\}_{k=0}^N$ with possibly non-uniform step sizes $\Delta t_k = t_{k+1} - t_k$.

SINDy assumes that $\mathbf{f}$ can be expressed as a sparse linear combination of candidate functions from a predefined library. Let $\mathbf{\Theta}(\mathbf{x}) = [\theta_1(\mathbf{x}), \ldots, \theta_P(\mathbf{x})]^\top \in \mathbb{R}^P$ denote the library evaluated at $\mathbf{x}$, and let $\mathbf{\Xi} \in \mathbb{R}^{P \times d}$ collect the unknown coefficients. The model class is

$$\mathbf{f}(\mathbf{x}) \approx \mathbf{\Theta}(\mathbf{x})^\top \mathbf{\Xi}. \tag{39}$$

If time derivatives $\dot{\mathbf{x}}(t_k)$ (or their estimates) of samples are available, SINDy identifies $\mathbf{\Xi}$ via sparsity-promoting regression:

$$\min_{\mathbf{\Xi}} \ \left\| \mathbf{\Theta}(\mathbf{X}) \, \mathbf{\Xi} - \dot{\mathbf{X}} \right\|_F^2 \ + \ \lambda \, \mathcal{R}(\mathbf{\Xi}), \tag{40}$$

where $\dot{\mathbf{X}} = [\dot{\mathbf{x}}(t_0)^\top; \ldots; \dot{\mathbf{x}}(t_N)^\top]$ and $\mathcal{R}(\cdot)$ enforces sparsity (e.g., an $\ell_0$-style penalty implemented by sequential thresholding, or an $\ell_1$ penalty). In practice, $\dot{\mathbf{X}}$ is often approximated from discrete timestamps; when $\Delta t_k$ is large, finite-difference estimates can become inaccurate, motivating derivative-free identification routes based on numerical integration constraints (e.g., Runge–Kutta formulations). However, when the time gaps become sufficiently large, even integration-based formulations may lose accuracy and lead to unstable or incorrect identification.

**SINDy with GFG** GFG is used as a *geometry-aware surrogate* for derivative estimation in sparse-sampling ODE discovery. In contrast to classical SINDy pipelines, we train the model using an on-manifold rollout loss implemented by integrating in the state latent space, and subsequently obtain tangent-consistent velocities via the Neural Tangent Projection. After training, the predicted velocities can be fed into a standard sparse regression (SINDy) procedure to recover an interpretable closed-form vector field.

Given paired observations $(x_k, x_{k+1})$ with a non-uniform step $\Delta t_k$, we obtain $(z_{s,k}, v_{\text{lat},k})$ by encoding $x_k$. To enable *state-dependent* latent dynamics while preserving the manifold constraint, we define a latent-space velocity predictor by decoding the current state latent and re-encoding its velocity:

$$\tilde{x}(z_s) = D_s(z_s), \qquad \mathcal{V}(z_s) = SoftVQ(E_v(\tilde{x}(z_s))), \tag{41}$$

where $D_s$ is the state decoder and $E_v$ denotes the velocity encoder. We then integrate the induced latent ODE

$$\dot{z}_s(t) = \mathcal{V}(z_s(t)), \tag{42}$$

using a one-step integrator

$$\hat{z}_{s,k+1} = \Phi_{\Delta t_k}(z_{s,k}; \mathcal{V}), \tag{43}$$

where $\Phi_{\Delta t_k}$ may query $\mathcal{V}$ at intermediate latent states. In our experiments, we use a second-order Runge–Kutta (midpoint) step:

$$k_1 = \mathcal{V}(z_{s,k}), \qquad k_2 = \mathcal{V}\big(z_{s,k} + \tfrac{\Delta t_k}{2} k_1\big), \qquad \hat{z}_{s,k+1} = z_{s,k} + \Delta t_k \, k_2. \tag{44}$$

We train GFG by matching the decoded one-step prediction to the observed next state:

$$\mathcal{L}_{task}^{physics} = \left\| D_s(\hat{z}_{s,k+1}) - \mathbf{x}_{k+1} \right\|_2^2, \tag{45}$$

where the loss is averaged over consecutive pairs in each mini-batch. This rollout objective directly supervises state transitions under the true (possibly non-uniform) time gaps $\Delta t_k$, without introducing an explicit candidate library or sparse-regression loss during representation learning. Together with the topological regularizer $\mathcal{L}_{\text{topo}}$ defined in the main text, the model is trained as a tangent-consistent velocity surrogate that respects the learned manifold geometry. After training, we obtain tangent velocities via the Neural Tangent Projection and apply the standard SINDy pipeline on the *training split* with a predefined library to recover a sparse, closed-form vector field; the recovered equation is then evaluated on the *test split* using one-step rollout MSE and VPT.

### C.2.2. MODEL HYPERPARAMETER SETTINGS

To reproduce the results in Table 2, we use the following configuration. We set the latent coordinate dimension to `coor_dim` $= 2$ and use an MLP backbone with hidden widths `hidden` $= (64, 32)$ for state encoder, velocity encoder and state decoder. We enable SoftVQ with `codebook_size` $= 32$, commitment weight `beta` $= 1.0$, temperature `tau` $= 1.0$, and VQ loss weight `w_vq` $= 0.5$. All experiments are run `seed` $= 42$.

### C.2.3. TRAINING AND POST-HOC EQUATION RECOVERY

**Training.** We train for `epochs` $= 1000$ using Adam with learning rate `lr` $= 0.01$ and batch size `BATCH_SIZE` $= 120$. We use the first $60\%$ of the trajectory as the training split (`TRAIN_FRAC` $= 0.6$). The loss weights are set to `loss_re` $= 1.0$, latent-rollout weight `w_rollout` $= 2.0$, and VQ weight `w_vq` $= 0.5$, matching the configuration used to generate the reported results.

**Post-hoc equation recovery.** After training, we compute observation-space velocities via the decoder JVP and perform sparse equation discovery on the training split using the given second-order polynomial library $\{x_1, x_2, x_1^2, x_1 x_2, x_2^2\}$. We fit coefficients using least squares followed by sequential thresholded least squares (STLS) with `max_iter` $= 10$. And we set the threshold $\tau = 0.1$.

## C.3. Case II: Kinetic Consistency for RNA Velocity

### C.3.1. IMPLEMENTATION

In the single-cell genomics regime, the dynamics are governed by the splicing kinetics of mRNA. For a specific gene, let $u$ denote unspliced counts and $s$ denote spliced counts. The theoretical velocity is given by the governing equations:

$$\frac{du}{dt} = \alpha - \beta u, \quad \frac{ds}{dt} = \beta u - \gamma s. \tag{46}$$

However, the kinetic rate parameters $\theta = [\alpha, \beta, \gamma]^\top$ are unknown and vary per gene. To enforce kinetic plausibility without explicitly learning these parameters for thousands of genes, we employ a projection-based residual loss.

We define a system matrix $M(u, s)$ that encapsulates the kinetic constraints. For a predicted velocity vector $v = [v_u, v_s]^\top$, we aim to find if there exists a valid parameter set $\theta$ such that $v$ is explained by the kinetics. We define:

$$M(u, s) = \begin{bmatrix} 1 & -u & 0 \\ 0 & u & -s \end{bmatrix}. \tag{47}$$

Ideally, the predicted velocity should lie in the column space of $M(u, s)$, i.e., $v = M(u, s)\theta$. We solve for the optimal implicit parameters $\theta^*$ via a closed-form ridge regression to handle numerical instability:

$$\theta^* = (M^\top M + \lambda_{reg} I)^{-1} M^\top v, \tag{48}$$

where $\lambda_{reg} = 10^{-2}$ is a regularization coefficient. The kinetic consistency loss is defined as the residual of this projection:

$$r = v - M(u, s)\theta^*, \quad \mathcal{L}_{task}^{bio} = \sum_{genes} \|r\|_2^2. \tag{49}$$

Intuitively, minimizing this residual projects the learned latent velocity onto the "Kinetic Subspace". A small residual implies that the direction inferred by our geometric flow is biologically plausible (i.e., it can be explained by some valid combination of transcription, splicing, and degradation rates), whereas a large residual indicates a violation of the underlying reaction kinetics.

C.3.2. DETAILS OF TRAINING

Prior to training, the single-cell RNA-seq data were preprocessed following the standard `scVelo` pipeline. Highly variable genes were selected by filtering and normalizing the data with a minimum shared count of 20, and the top 2,000 genes with the highest variability were retained. Principal component analysis (PCA) was then applied to reduce the data to 30 dimensions, and a neighborhood graph was constructed using 20 nearest neighbors in the PCA space. First- and second-order moments were computed for velocity estimation, and UMAP embeddings were generated for visualization.

The model encodes genes into a latent dimension of 8 using an encoder consisting of four hidden layers with sizes 256, 512, 512, and 256. The latent representations are then quantized using a vector quantization (VQ) codebook containing 32 codewords of dimension 8, matching the encoder output. A decoder, symmetric to the encoder, reconstructs the gene expression profiles from the quantized latent vectors. All model components are trained jointly, with parameters updated using the Adam optimizer. The model is trained for 10 epochs with a batch size of 128 and an initial learning rate of $1 \times 10^{-3}$.

The training objective is a weighted combination of multiple losses. We adjust their weights to ensure the balance of losses:

- ODE loss: $\text{loss}_{\text{ode}}$ with weight 20
- Spliced MSE loss: $\text{loss}_s$ with weight 2.5
- Unspliced MSE loss: $\text{loss}_u$ with weight 7.5
- Vector quantization losses: $\text{loss}_{m\_vq}$ and $\text{loss}_{v\_vq}$, each with weight 1
- Smoothness regularization: $\text{loss}_{\text{smooth}}$ with weight 300
- Alignment constraint: $\text{loss}_{\text{align}}$ with weight 300

The model is optimized using the Adam optimizer (or equivalent) to update parameters according to the gradient of the weighted loss function, enabling learning of both the underlying dynamics and the latent representation.

C.3.3. LOSS FUNCTIONS

The training objective is a weighted combination of several loss terms. Below we summarize their core formulas.

**Spliced and Unspliced MSE Loss.** For predicted velocities or expression values $\hat{x}$ and ground truth $x$, the mean squared error (MSE) is used:

$$\mathcal{L}_{s/u} = \frac{1}{B \cdot G} \sum_{i=1}^{B} \sum_{g=1}^{G} (\hat{x}_{i,g} - x_{i,g})^2$$

**Vector Quantization Loss.** Given a set of input vectors $z$ and a learnable codebook $E$, we compute soft assignments

$$p_i = \text{softmax}\left( - \frac{\|z_i - E_k\|^2}{\tau} \right) \quad \text{or} \quad p_i = \text{softmax}\left( \frac{z_i^\top E_k}{\tau} \right)$$

depending on whether Euclidean or cosine similarity is used. The quantized vectors are

$$z_i^q = \sum_k p_{ik} E_k$$

The VQ loss combines a commitment term and an entropy regularization:

$$\mathcal{L}_{\text{commit}} = \|z - z_{\text{detach}}^q\|_2^2, \quad \mathcal{L}_{\text{entropy}} = \log K + \sum_{k=1}^{K} \bar{p}_k \log \bar{p}_k, \quad \bar{p}_k = \frac{1}{N} \sum_i p_{ik}$$

The final VQ loss is

$$\mathcal{L}_{\text{VQ}} = \beta \, \mathcal{L}_{\text{commit}} + \gamma \, \mathcal{L}_{\text{entropy}}$$

**Smoothness Regularization.** Let $v_i$ be the predicted velocity for cell $i$ and $A$ the adjacency matrix; we encourage similar velocities in neighbors:

$$\mathcal{L}_{\text{smooth}} = \frac{\sum_{i,j} A_{ij}\big(1 - \cos(v_i, v_j)\big)}{\sum_{i,j} A_{ij}}$$

**Alignment Loss.** Because of the natural orthogonality of vectors in high dimension, we choose to do alignment in low dimension space. Firstly we project the expression $x$ and velocity $v$ onto top $d$ principal components:

$$x^{\text{PCA}} = xV, \quad v^{\text{PCA}} = vV, \quad V \in \mathbb{R}^{G \times d}$$

For each edge $(i, j)$ in adjacency $A$, align predicted velocity to expression change:

$$\mathcal{L}_{\text{align}} = \frac{1}{|E|} \sum_{(i,j)\in E} \Big[1 - \cos(v_i^{\text{PCA}}, x_j^{\text{PCA}} - x_i^{\text{PCA}})\Big]$$

**ODE Residual Loss.** Following the derivation in Section C.3.1, we define the residual and the corresponding task loss as

$$r = v - M(u, s)\theta^*, \qquad \mathcal{L}_{\text{task}}^{\text{bio}} = \sum_{\text{genes}} \|r\|_2^2.$$

## C.4. Case III: AI-Video Detection

### C.4.1. IMPLEMENTATION

**Feature Extraction and Representation** We extract the features based on second-order temporal differences. For an input video clip, we consider the first $L = 10$ frames and project them into the latent space of Stable Diffusion v1.5 (Rombach et al., 2022) via the VAE encoder, obtaining the latent sequence $\mathcal{X} = \{X_i\}_{i=1}^{L}$. For each latent $X_i$, we define its generative score direction $v_i$ by performing a one-step reverse diffusion at a minimal timestep $t = 0.005$. Let $\epsilon_\theta(\cdot)$ be the pre-trained UNet denoiser; the score direction is defined as the negative predicted noise:

$$v_i = -\epsilon_\theta(X_i + \epsilon, t) \tag{50}$$

where $\epsilon$ is a small Gaussian perturbation. This vector $v_i$ represents the model's preferred update direction in the latent manifold. To map the latent scores back to the image manifold, we compute the Jacobian-Vector Product (JVP) of the VAE decoder $\mathcal{G}(\cdot)$. This represents the first-order change in the reconstructed image given a perturbation in the latent score direction. For each frame, the predicted temporal variation $\delta_{i,\text{pred}}$ is derived as:

$$\delta_{i,\text{pred}} = \nabla_X \mathcal{G}(X_i) \cdot v_i \tag{51}$$

Simultaneously, the observed temporal variation $\delta_{i,\text{obs}}$ is defined as the actual inter-frame transition, computed via the first-order forward difference between consecutive reconstructed frames:

$$\delta_{i,\text{obs}} = \mathcal{G}(X_{i+1}) - \mathcal{G}(X_i) \tag{52}$$

To isolate the high-order artifacts, we compute the predicted acceleration $\Delta^2 X_{i,\text{pred}}$ and observed acceleration $\Delta^2 X_{i,\text{obs}}$ by applying a second-order difference operator to the respective velocity sequences. For each frame index $i \in \{1, \ldots, 8\}$, the accelerations are derived as:

$$\Delta^2 X_{i,\text{obs}} = \frac{\delta_{i+1,\text{obs}} - \delta_{i,\text{obs}}}{h} \approx \frac{\mathcal{G}(X_{i+2}) - 2\mathcal{G}(X_{i+1}) + \mathcal{G}(X_i)}{h^2} \tag{53}$$

The final feature $F_i$ is obtained by the pixel-wise difference between the predicted variation and the observed transition:

$$F_i = \frac{(\Delta_{i,\text{pred}} - \Delta_{i,\text{obs}})^2}{\Delta i, \text{obs}^2 + \eta} \tag{54}$$

For a 10-frame input, this yields a sequence of $T = 8$ high-order maps $F \in \mathbb{R}^{T \times 3 \times H \times W}$.

To evaluate both generative artifacts and dynamic inconsistencies, we build upon the NSG-VD (Zhang et al., 2025) framework and implement a dual-stream fusion strategy. We extract the NSG features (3 channels) from the temporal layers of the pre-trained SVD model, denoted as $\mathcal{F}_{\text{nsg}} \in \mathbb{R}^{T \times 3 \times H \times W}$. Both the NSG feature stream ($\mathcal{F}_{\text{nsg}}$) and our feature stream (denoted here as $\mathcal{F}_{\text{kin}} = F$) are independently projected into a latent manifold via its own discriminator for Deep MMD training, producing per-video scores $s_{\text{nsg}}$ and $s_{\text{kin}}$. The final detection score is obtained by equal-weighted fusion after z-score normalization on the real reference set:

$$s_{\text{fused}} = \tfrac{1}{2}\, \tilde{s}_{\text{nsg}} + \tfrac{1}{2}\, \tilde{s}_{\text{kin}} \tag{55}$$

where $\tilde{s}_{(\cdot)}$ denotes the z-score normalized MMD score.

**Deep MMD Scoring Process**  The detection score is derived by measuring the distribution distance between a test video $X_{test}$ and a set of pristine (real) reference videos $\mathcal{X}_{ref} = \{x_i\}_{i=1}^{N}$. We employ the Maximum Mean Discrepancy (MMD) in the learned latent space. Given the mapping function $\phi$, the Deep MMD score $S$ is computed as:

$$S = \left| \frac{1}{N} \sum_{i=1}^{N} k(\phi(x_i), \cdot) - k(\phi(X_{test}), \cdot) \right|_{\mathcal{H}}^2, \tag{56}$$

where $k(\cdot, \cdot)$ is a learned Gaussian kernel $k(z, z') = \exp(-\frac{\|\phi(z) - \phi(z')\|^2}{\sigma^2})$.

To be more specific, we expand the squared norm into three accessible terms:

$$S = \underbrace{\frac{1}{N^2} \sum_{i=1}^{N} \sum_{j=1}^{N} k(\phi(x_i), \phi(x_j))}_{\mathcal{T}_1:\text{Reference Cohesion}} - \underbrace{\frac{2}{N} \sum_{i=1}^{N} k(\phi(x_i), \phi(X_{test}))}_{\mathcal{T}_2:\text{Cross-Similarity}} + \underbrace{k(\phi(X_{test}), \phi(X_{test}))}_{\mathcal{T}_3:\text{Self-Similarity}}. \tag{57}$$

Here, $\mathcal{T}_1$ represents the internal consistency of the authentic video distribution, which acts as a baseline constant. $\mathcal{T}_2$ measures the affinity between the test video and the authentic manifold.

During training, the training loss is defined as:

$$\mathcal{L}MMD = -\frac{\widehat{\text{MMD}}^2(\mathcal{X}re, \mathcal{X}_{fa})}{\sqrt{\text{Var}[\widehat{\text{MMD}}^2] + \eta}}, \tag{58}$$

where $\widehat{\text{MMD}}^2$ is the empirical estimator computed similarly to the expansion in the scoring process, but applied symmetrically between the real and fake batches. The denominator $\sqrt{\text{Var}[\cdot]}$ represents the standard deviation of the MMD estimator.

### C.4.2. DETAILS OF TRAINING

- **Kernel:** The kernel bandwidth $\sigma$ and sensitivity parameters $\epsilon, \sigma_0$ are initialized at $1000.0, 10.0$, and $0.1$, respectively.

- **Optimizer Settings:** AdamW optimizer, learning rate $= 1 \times 10^{-4}$.

- **Batch Size:** 64.

- **Random Seed:** 42.

- **Training Duration:** 200 epochs.

- **Input Resolution:** $8 \times 224 \times 224$ (T $\times$ H $\times$ W).

# D. Supplementary Experiments and Analysis

### D.1. Case I: ODE Discovery

We further evaluate GFG on more challenging dynamical systems beyond the 2D circular-motion benchmark, including non-autonomous ODEs, nonlinear Hamiltonian dynamics, and a PDE-scale Burgers' equation. These experiments test whether the proposed tangent-grounded latent dynamics remain effective under time-dependent, nonlinear, and high-dimensional settings.

**Time-dependent vector fields.** GFG can be naturally extended to non-autonomous systems by conditioning the latent velocity predictor on time through positional embeddings, while keeping the tangent projection mechanism unchanged. We validate this extension on a time-varying circle-rotation ODE with a time-augmented SINDy library. As shown in Table 7, GFG achieves substantially lower rollout error and longer valid prediction time, and is the only method that recovers the correct time-dependent sparse structure.

| Method | MSE ↓ | VPT ↑ |
|---|---|---|
| SINDy | 1.289 | 0.23 |
| IRK-SINDy | 1.187 | 0.17 |
| **GFG (Ours)** | **0.068** | **2.16** |

*Table 7.* **Evaluation on a time-dependent ODE.**

**Nonlinear Hamiltonian dynamics.** We next evaluate GFG on a 3D nonlinear Hamiltonian system under the same sparse, large-time-interval setting. The ground-truth system is

$$\dot{X} = -6Y\sin(6Z), \quad \dot{Y} = 6X\sin(6Z) - 6Z\sin(6X), \quad \dot{Z} = 6Y\sin(6X).$$

As shown in Table 8, GFG obtains the best MSE and VPT, and is the only method that recovers the correct sparse term structure, although the estimated coefficients remain attenuated under sparse observations.

| Method | MSE ↓ | VPT ↑ | Discovered ODE |
|---|---|---|---|
| Ground Truth | – | – | $\dot{X} = -6.000Y\sin(6Z)$; $\dot{Y} = 6.000X\sin(6Z) - 6.000Z\sin(6X)$; $\dot{Z} = 6.000Y\sin(6X)$ |
| SINDy | 0.1100 | 0.0167 | $\dot{X} = 0.624Z\sin(6X)$; $\dot{Y} = -0.861Y\sin(6Z) - 0.940Y\sin(6X)$; $\dot{Z} = 0.649X\sin(6Z)$ |
| IRK-SINDy | 0.1117 | 0.0132 | $\dot{X} = 1.979X\sin(6Z) + 2.406Z\sin(6X) + 1.575Y\sin(6X)$; $\dot{Y} = -3.634Y\sin(6Z)$; $\dot{Z} = 4.160X\sin(6Z) - 1.950Z\sin(6X)$ |
| MNN | 0.1229 | 0.0184 | $\dot{X} = 0.028X\sin(6Z) + 0.107Z\sin(6X) + 0.779Y\sin(6X)$; $\dot{Y} = 0.449Y\sin(6Z) - 1.942Z\sin(6X)$; $\dot{Z} = -0.035X\sin(6Z) + 0.015Z\sin(6X) - 0.610Y\sin(6X)$ |
| S-MNN | 0.1398 | 0.0153 | $\dot{X} = 0.361Y\sin(6Z)$; $\dot{Y} = -0.507X\sin(6Z) + 0.506Z\sin(6X)$; $\dot{Z} = -0.360Y\sin(6X)$ |
| **GFG (Ours)** | **0.0420** | **0.0310** | $\dot{X} = -2.021Y\sin(6Z)$; $\dot{Y} = 4.363X\sin(6Z) - 4.146Z\sin(6X)$; $\dot{Z} = 2.013Y\sin(6X)$ |

*Table 8.* **Evaluation on a 3D nonlinear Hamiltonian system.** GFG achieves the best rollout accuracy and valid prediction time, and recovers the correct sparse term structure under large time intervals.

**PDE-scale dynamics.** We also extend GFG to Burgers' equation on a 256-point spatial grid. After training, the learned latent-space velocities are projected onto a PDE candidate library and fitted by sparse regression. GFG recovers

$$u_t = 0.091u_{xx} - 0.991uu_x,$$

which matches the correct sparse PDE structure. This result suggests that GFG can be applied beyond low-dimensional ODE discovery to PDE-scale dynamical systems, where integral-regression methods such as IRK-SINDy become computationally expensive due to their cubic scaling.

### D.1.1. SCALABILITY COMPARISON WITH GEOMETRY-AWARE METHODS

We further compare GFG with representative geometry-aware continuous-time models. We use Neural Manifold ODE (NM-ODE; Lou et al. 2020) as a chart-based reference and Riemannian Flow Matching (RFM; Chen & Lipman 2024) as a flow-matching reference. These methods are included as geometric reference points rather than fully task-matched competitors. NM-ODE and RFM typically assume known or analytically specified manifold geometry, whereas GFG learns

| $D$ | NM-ODE (Lou et al., 2020) | | | | RFM (Chen & Lipman, 2024) | | | | GFG (Ours) | | | |
|---|---|---|---|---|---|---|---|---|---|---|---|---|
| | MSE ↓ | Params | Time ↓ (ms) | FLOPs ↓ | MSE ↓ | Params | Time ↓ (ms) | FLOPs ↓ | MSE ↓ | Params | Time ↓ (ms) | FLOPs ↓ |
| 2 | 1.21 | 33.8K | 668.28 | ∼1.34M | 3.23 | 53.3K | 139.45 | ∼1.89M | **0.23** | 53.9K | **14.04** | ∼**208.4K** |
| 16 | 0.94 | 37.4K | 3,424.16 | ∼1.50M | 2.05 | 61.5K | 267.45 | ∼3.73M | **0.21** | 61.0K | **14.34** | ∼**236.8K** |
| 64 | 1.00 | 49.7K | 13,435.96 | ∼2.17M | 1.72 | 92.4K | 945.25 | ∼20.28M | **0.23** | 89.5K | **14.06** | ∼**352.0K** |
| 128 | 1.44 | 66.2K | 27,021.86 | ∼3.36M | 2.65 | 133.6K | 1,904.95 | ∼61.78M | **0.33** | 129.5K | **17.51** | ∼**518.1K** |

*Table 9.* **Controlled scalability comparison with representative geometry-aware methods.** We evaluate NM-ODE, RFM, and GFG on a $D$-dimensional harmonic oscillator with states on $(S^1)^{D/2}$ using one-step prediction. NM-ODE and RFM serve as geometric reference points rather than fully task-matched competitors. GFG achieves substantially lower runtime and FLOPs across dimensions by constructing tangent velocities with a single matrix-free JVP, while avoiding explicit chart construction or full Jacobian-based projection.

the manifold directly from data. Moreover, they couple geometry and dynamics within a single modeling objective, while GFG explicitly decouples topological state learning from velocity primitive modeling.

We conduct a controlled experiment on a $D$-dimensional harmonic oscillator with states on $(S^1)^{D/2}$, using one-step prediction $x_k \rightarrow x_{k+1}$. Each method is trained with its canonical objective. NM-ODE uses a CNF likelihood objective and requires sequential autograd calls for divergence estimation. RFM uses flow matching with tangent projection, which involves Jacobian-based projection operations. In contrast, GFG constructs the tangent velocity with a single matrix-free JVP, $J_G(z)v_{\text{lat}}$, avoiding explicit chart construction or full Jacobian-based projection. Results are shown in Table 9

NM-ODE exhibits super-linear runtime growth due to sequential autograd overhead in divergence computation. RFM becomes increasingly expensive as the ambient dimension grows, due to repeated Jacobian-based tangent projections. GFG remains substantially more efficient because its tangent construction only requires a single matrix-free JVP, which is efficiently parallelized. These results support our scalability claim in learned high-dimensional observational manifolds.

### D.2. Case II: Single-Cell RNA Velocity

#### D.2.1. DETAILED INTERPRETABILITY ANALYSIS ON MOUSE BRAIN

In this section, we provide a comprehensive quantitative validation of the lineage disentanglement capabilities discussed in the main text. To ensure an unbiased analysis, we first established a data-driven protocol to identify the representative codebook atoms for each lineage. Specifically, we calculated the mean attention weight of all atoms across the annotated cell clusters. The *Atom 4* and *Atom 11* are shown as examples. *Atom 4* emerged as the top-ranking atom within the neuronal clusters (including V-SVZ, IPC, and cortical layers), while *Atom 11* exhibited the highest mean activation specifically within the glial clusters (RG, Astro, OPC). Accordingly, we selected these two atoms as the representative dynamic motifs for the neuronal and glial lineages, respectively.

We then decoded these atoms into gene expression space to verify their biological semantics via Gene Ontology (GO) enrichment analysis. As illustrated in Figure 7, the functional profiles of these two atoms exhibit a sharp dichotomy. The neuronal atom (#4) is strictly associated with neurogenesis-related processes, showing high significance for terms such as *nervous system development* ($p < 10^{-5}$) and *synapse organization*. Conversely, the glial atom (#11) enriches for *extracellular matrix organization* and *regulation of cell differentiation*. These functional terms align precisely with the distinct biological roles of the two lineages: signal processing and neurite extension for neurons, versus structural scaffolding and myelination for glial cells.

This semantic separation is quantitatively corroborated by the cluster-wise activation heatmap shown in Figure 8. The heatmap reveals that Atom 11 functions as a highly specialized branching module, displaying strong activation exclusively within the Glial clusters (highlighted in the blue box) while remaining suppressed across the neuronal trajectory. In contrast, Atom 4 demonstrates a broad activation pattern that covers the entire neurogenic path. Finally, an inspection of the top-weighted molecular drivers further validates these findings. The neuronal atom is driven by canonical markers such as *Tubb3*, *Stmn2*, and *Neurod1*, whereas the glial atom is dominated by master regulators of oligodendrocyte specification, including *Olig1*, *Olig2*, and *Pdgfra*. Together, these results confirm that GFG autonomously learns to orthogonally represent competing cell fates through discrete geometric flows.

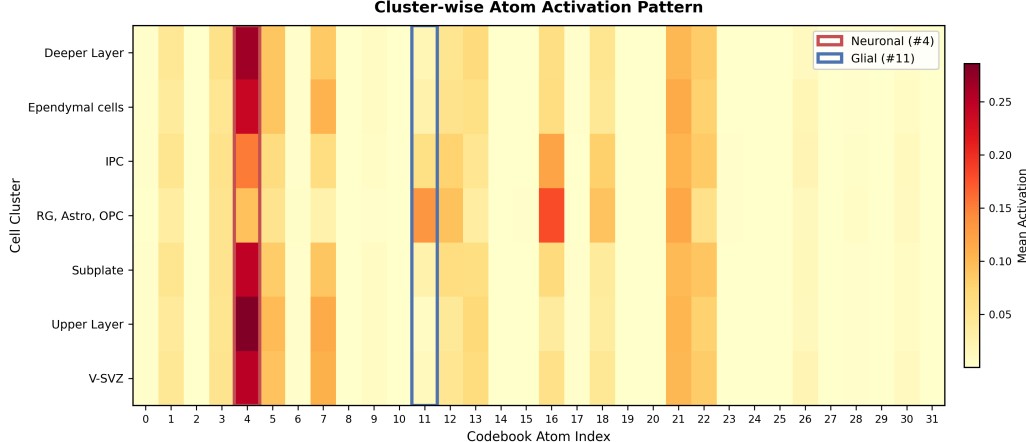

*Figure 8.* **Lineage specificity.** Atom 4 (red) is broadly neuronal; Atom 11 (blue) is specific glial.

## D.3. Case III: AI-Video Detection

**Detailed Quantitative Comparison** In this section, we present the comprehensive quantitative breakdown across all ten individual video generation benchmarks to further substantiate the zero-shot generalization capabilities of GFG reported in the main text(as shown in Table 10).

**Robustness under Observation Degradation** Real-world dynamical observations are often affected by noise, measurement errors, and perceptual degradation. To evaluate whether the proposed tangent projection residual is robust to such perturbations, we conduct an additional experiment on AI-generated video detection by progressively degrading the input resolution. Specifically, we downsample each video with compression rates $R_1 = 2$ and $R_2 = 4$, and then evaluate the detection performance under the same protocol. As shown in Table 11, the performance of NSG-VD and NPR drops substantially after compression, indicating that purely statistical or artifact-based signals can be sensitive to observation quality. In contrast, GFG + NSG-VD maintains consistently high AUROC across all compression levels, with 96.3% on the original videos, 97.0% under $R_1 = 2$, and 96.9% under

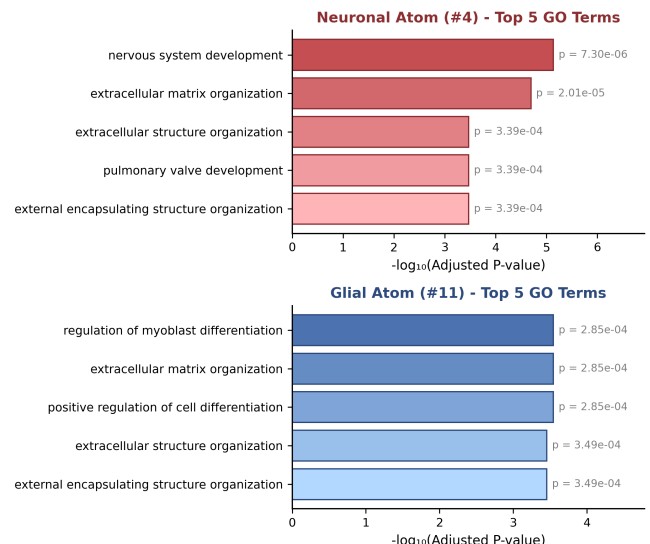

*Figure 7.* **Functional verification.** Top: Neuronal atom (#4) enriches for neurogenesis. Bottom: Glial atom (#11) enriches for structure.

$R_2 = 4$. This stability suggests that the tangent projection residual is less dependent on high-frequency pixel details and instead captures more intrinsic geometric inconsistencies in the temporal evolution.

## E. Detailed Case-specific Related Work

### E.1. RNA Velocity Inference

scVelo (Bergen et al., 2020): The standard baseline that generalizes RNA velocity by solving the full transcriptional dynamics (induction, repression, steady states) using an Expectation-Maximization (EM) algorithm. veloVI(Gayoso et al., 2024): A statistical framework (often using variational inference or non-linear dynamics) that introduces a unified latent time variable to resolve complex lineage kinetics and multi-furcations. DeepVelo (Cui et al., 2024): A deep learning method using Graph Convolutional Networks (GCNs) to predict cell-specific splicing kinetics, allowing for time-dependent rates rather than constant gene-specific parameters. TIvelo (Ge et al., 2025): A recent approach that first infers coarse-grained trajectory directions at the cluster level (using TI methods) to guide the fine-grained cell-level velocity estimation, avoiding

| Method | Metric | M.Scope | M.Studio | M.Valley | HotShot | Show1 | Gen2 | Crafter | Lavie | Sora | Wild | Avg. |
|---|---|---|---|---|---|---|---|---|---|---|---|---|
| **DeMamba** | Recall | 87.00 | 93.60 | 98.80 | 40.60 | 48.40 | 98.00 | 88.40 | 59.00 | 48.21 | 58.20 | 72.02 |
| | Accuracy | 91.70 | 95.00 | 97.60 | 68.50 | 72.40 | 97.20 | 92.40 | 77.70 | 72.32 | 77.30 | 84.21 |
| | F1 | 91.29 | 94.93 | 97.63 | 56.31 | 63.68 | 97.22 | 92.08 | 72.57 | 63.53 | 71.94 | 80.12 |
| | AUROC | 98.04 | 98.82 | 99.68 | 87.84 | 90.12 | 99.46 | 97.81 | 91.32 | 88.36 | 87.38 | 93.88 |
| **NPR** | Recall | 61.20 | 80.00 | 98.00 | 16.00 | 33.00 | 91.20 | 80.60 | 34.60 | 35.71 | 43.20 | 57.35 |
| | Accuracy | 79.80 | 89.20 | 98.20 | 57.20 | 65.70 | 94.80 | 89.50 | 66.50 | 67.86 | 70.80 | 77.96 |
| | F1 | 75.18 | 88.11 | 98.20 | 27.21 | 49.03 | 94.61 | 88.47 | 50.81 | 52.63 | 59.67 | 68.39 |
| | AUROC | 93.05 | 97.18 | 99.66 | 82.97 | 90.50 | 99.13 | 97.87 | 87.54 | 90.47 | 91.84 | 93.02 |
| **TALL** | Recall | 51.20 | 65.20 | 93.40 | 32.00 | 61.60 | 94.80 | 81.80 | 49.20 | 25.00 | 53.60 | 60.78 |
| | Accuracy | 75.10 | 82.10 | 96.20 | 65.50 | 80.30 | 96.90 | 90.40 | 74.10 | 61.61 | 76.30 | 79.85 |
| | F1 | 67.28 | 78.46 | 96.09 | 48.12 | 75.77 | 96.83 | 89.50 | 65.51 | 39.44 | 69.34 | 72.63 |
| | AUROC | 95.82 | 97.14 | 99.73 | 92.55 | 97.36 | 99.79 | 99.09 | 94.84 | 86.67 | 93.75 | 95.67 |
| **STIL** | Recall | 73.80 | 70.80 | 43.40 | 1.40 | 2.00 | 45.00 | 13.20 | 7.20 | 1.79 | 11.60 | 27.02 |
| | Accuracy | 86.90 | 85.40 | 71.70 | 50.70 | 51.00 | 72.50 | 56.60 | 53.60 | 50.89 | 55.80 | 63.51 |
| | F1 | 84.93 | 82.90 | 60.53 | 2.76 | 3.92 | 62.07 | 23.32 | 13.43 | 3.51 | 20.79 | 35.82 |
| | AUROC | 96.43 | 97.77 | 99.34 | 86.66 | 90.56 | 98.88 | 97.04 | 88.16 | 92.57 | 87.52 | 93.49 |
| **NSG-VD** | Recall | 68.33 | 98.33 | 100.0 | 92.50 | 87.50 | 80.00 | 98.33 | 94.17 | 78.57 | 82.50 | 88.02 |
| | Accuracy | 81.67 | 98.33 | 96.67 | 91.67 | 90.83 | 88.33 | 95.83 | 94.17 | 88.39 | 88.75 | 91.46 |
| | F1 | 78.85 | 98.33 | 96.77 | 91.74 | 90.52 | 87.27 | 95.93 | 94.17 | 87.13 | 88.00 | **90.87** |
| | AUROC | 92.26 | 98.66 | 98.15 | 94.45 | 96.38 | 94.83 | 98.16 | 97.41 | 96.40 | 94.73 | 96.14 |
| **GFG(Ours)** | Recall | 64.14 | 51.52 | 100.00 | 48.85 | 89.31 | 91.43 | 71.01 | 85.27 | 66.92 | 67.81 | 73.63 |
| | Accuracy | 79.73 | 74.41 | 98.81 | 72.16 | 96.61 | 95.25 | 82.03 | 95.64 | 99.72 | 87.27 | 88.16 |
| | F1 | 76.20 | 67.68 | 99.09 | 61.38 | 97.58 | 95.41 | 76.84 | 97.46 | 62.76 | 80.65 | 81.50 |
| | AUROC | 95.50 | 94.77 | 99.34 | 91.74 | 97.29 | 97.12 | 97.16 | 93.14 | 96.94 | 70.93 | 93.39 |
| **GFG + NSG-VD** | Recall | 81.93 | 82.30 | 100.00 | 89.05 | 100.00 | 98.13 | 100.00 | 100.00 | 84.42 | 96.57 | **93.24** |
| | Accuracy | 85.59 | 85.05 | 91.63 | 89.06 | 99.35 | 93.39 | 92.43 | 100.00 | 91.16 | 94.42 | **92.21** |
| | F1 | 84.79 | 84.69 | 91.51 | 87.97 | 100.00 | 94.28 | 95.07 | 100.00 | 53.65 | 96.63 | 88.86 |
| | AUROC | 95.59 | 95.75 | 99.29 | 91.78 | 98.19 | 97.97 | 97.33 | 95.83 | 96.40 | 94.50 | **96.26** |

*Table 10.* **Comprehensive Quantitative Comparisons on Zero-Shot Video Verification.** We report the performance across ten diverse benchmarks and the overall average. GFG (Ours)+NSG-VD demonstrates superior robustness, achieving the highest average performance across most metrics (Recall, Accuracy, AUROC).

local direction inconsistencies. Graphvelo (Chen et al., 2025b): A geometric method that takes existing velocity outputs (from other methods) and projects them onto the tangent space of the cell manifold (approximated by kNN) to strictly enforce geometric consistency.

### E.2. AI-Video Detection

NSG-VD (Zhang et al., 2025): A physics-driven paradigm that quantifies a novel statistic called Normalized Spatiotemporal Gradient (NSG). By leveraging pre-trained diffusion models to estimate probability flow gradients, it explicitly detects the violations of physical conservation laws. D3 (Zheng et al., 2025): A training-free framework that detects AI-generated content by analyzing second-order statistical features. It reveals subtle spatiotemporal inconsistencies, providing an adaptable solution for zero-shot detection. STIL (Gu et al., 2021): A spatiotemporal incoherent learning framework that slices video volumes into 2D maps to capture subtle dynamic inconsistencies across frames, without the heavy computational cost of 3D convolutions. TALL (Xu et al., 2023): A temporal-aware approach that rearranges consecutive frames into a predefined thumbnail layout for efficient spatiotemporal modeling. DeMamba (Chen et al., 2024): A large-scale detection framework that establishes the million-scale GenVideo benchmark and utilizes a Mamba-based architecture to resolve long-range dependencies. DIVID (Liu et al., 2024): A detection framework that utilizes a CNN-LSTM architecture to jointly process RGB frames and Diffusion Reconstruction Error (DIRE) across multiple frames. NPR (Tan et al., 2024): A study

| Method | Original | $R_1 = 2$ | $R_2 = 4$ |
|---|---|---|---|
| NSG-VD | 96.1 | 66.0 | 65.8 |
| NPR | 93.0 | 79.9 | 67.1 |
| **GFG + NSG-VD** | **96.3** | **97.0** | **96.9** |

*Table 11.* **Robustness under observation degradation.** We report AUROC under increasing compression by downsampling the input resolution. GFG + NSG-VD remains stable under strong degradation, suggesting that the tangent projection residual captures intrinsic geometric inconsistencies rather than fragile low-level pixel artifacts.

that rethinks up-sampling operations in CNN-based generators to identify universal artifacts for generalizable deepfake detection. VidGuard (Park et al., 2025): A reasoning-based framework that leverages multi-modal large language models and reinforcement learning to provide both detection results and interpretative explanations. Optical Flow based CNN (Amerini et al., 2019): A motion-based detection method that utilizes optical flow fields as input to a CNN to model temporal inconsistencies. AltFreezing (Wang et al., 2023): A generalizable detection strategy that introduces an alternating freezing mechanism during training to balance spatial and temporal feature learning. More broadly, recent studies on world models and geometric reasoning also emphasize the importance of consistency as a core principle. Trinity of Consistency (Wei et al., 2026) identifies modal, spatial, and temporal consistency as defining properties for general world models. Faire (Zhang et al., 2026) studies how reinforcement learning can improve functional alignment in geometric interleaved reasoning, providing a complementary perspective on using geometric structure to guide reliable model behavior.

