# OpenReview forum: "Geometric Flow Grounding: A Unified Manifold Decoupling Framework for Dynamics Discovery and Verification"
_ICML.cc/2026/Conference — ICML 2026 spotlight_

### Official Review · Reviewer_UNwE · 2026-03-08

**Soundness:** 2
**Presentation:** 3
**Significance:** 3
**Originality:** 3
**Overall Recommendation:** 4
**Confidence:** 4

**Summary:**

The paper studies the problem of modeling dynamical systems from high-dimensional observations while respecting underlying geometric structure. Building on Neural ODE-style models, the authors argue that learning dynamics directly in ambient Euclidean space can lead to off-manifold drift and accumulated integration errors. To address this, the paper proposes Geometric Flow Grounding, which assumes that the data lie on a lower-dimensional manifold parameterized by a learned generator $G_\theta$. The method constrains the learned vector field to lie in the corresponding tangent space defined by the decoder Jacobian and introduces a Neural Tangent Projection Layer to enforce this constraint efficiently using Jacobian–vector products. Experiments on synthetic and real-world datasets demonstrate the proposed method’s potential advantages.

**Compliance With Llm Reviewing Policy:**

Affirmed.

**Final Justification:**

The paper proposes a novel geometry-aware framework for learning dynamical systems using a learned manifold and tangent-space dynamics. The idea is original and well-motivated, and the empirical results are promising.

The rebuttal substantially improves the paper. In particular, the authors provide additional ablations and experiments, and importantly, clarify the computational aspects of the method. The added comparisons and complexity analysis address my previous concerns regarding scalability relative to geometry-based approaches.

Regarding soundness, the authors now make the assumptions underlying the learned manifold and tangent-space construction more explicit and provide a robustness argument. While this improves the clarity and theoretical grounding, I still have some reservations about how well these assumptions hold in complex real-world settings and how sensitive the method is to violations of them.

Overall, the paper has improved meaningfully after rebuttal. While some concerns remain, I find the contribution sufficiently interesting and better supported, and I therefore raise my score to weak accept.

**Key Questions For Authors:**

(1) The problem formulation assumes a time-invariant vector field $v^{\ast}(x)$. Many real-world systems exhibit time-dependent dynamics. Could the authors clarify whether the proposed framework can be extended to handle time-dependent vector fields $v^{\ast}(x,t)$, and whether such an extension would require significant modifications to the current approach?

(2) Real-world dynamical systems are often affected by noise, measurement errors, or stochastic fluctuations. The manuscript primarily evaluates the method on deterministic settings. Could the authors discuss the robustness of the proposed approach under noisy observations or stochastic dynamics?

**Limitations:**

The manuscript does not explicitly discuss methodological limitations of the proposed framework. For instance, the approach relies on a learned generator $G_\theta$ to parameterize the data manifold, and the validity of the tangent-space construction depends on the quality of this representation. Errors in the learned manifold or its Jacobian may therefore affect the accuracy of the projected dynamics. Additionally, constraining the vector field to the learned tangent space may limit the expressiveness of the model class when the manifold approximation is imperfect. A discussion of such limitations would strengthen the manuscript.

Furthermore, the impact statement focuses primarily on potential positive applications. It would benefit from a more balanced discussion of possible limitations and risks. In particular, since the proposed approach relies on learned approximations of geometric structure, predictions may still be subject to modeling errors and should be carefully validated when applied in sensitive decision-making contexts.

**Strengths And Weaknesses:**

### Strengths
- The paper is well-structured and clearly written. The proposed approach appears novel and addresses an important challenge in modeling dynamical systems, particularly learning dynamics from high-dimensional observations while respecting geometric constraints.

- The introductory figures effectively support the reader’s understanding of the proposed framework.

- The divide-and-conquer approach used to separate state representation and motion information is an interesting design choice that naturally encourages the model to respect geometric constraints.

- The synthetic experiment on the 1D unit sphere provides a clear proof-of-concept demonstrating that the proposed projection mechanism improves learning from sparsely sampled trajectories. Additionally, experiments on four scRNA-seq datasets with richer intrinsic manifold structure are presented, including comparisons with five state-of-the-art methods using three evaluation metrics. The reported results show that Geometric Flow Grounding outperforms the considered baselines on these datasets.

- The idea of repurposing the geometric projection residual as a zero-shot metric for deepfake video detection is interesting. The results reported in Table 4 suggest that Geometric Flow Grounding can provide a useful auxiliary signal that may improve the robustness of existing detection methods.

### Weaknesses
- The manuscript assumes that the data manifold is parameterized by a learned generator $G_\theta$, whose Jacobian is used to define the tangent space. While the proposed physics-based constraint functional (implemented via the VQ-VAE framework) encourages the learned representation to respect dynamical structure, the resulting manifold geometry remains induced by the learned model. Consequently, the validity of the tangent space construction still relies on implicit assumptions such as smoothness and locally constant rank of $G_\theta$, which are not explicitly discussed.

- The implementation details are somewhat under-specified. In particular, the main text does not clearly describe the model architectures used, and the supplementary material only briefly outlines them. As a result, it is difficult to reconstruct the full implementation. Providing more detailed architectural descriptions and releasing code would improve reproducibility.

- The manuscript argues that existing approaches in geometric deep learning rely on computationally expensive operations (e.g., local coordinate charts or optimal transport) that limit scalability in high-dimensional settings. However, the paper does not provide comparisons to such approaches, either empirically or through a complexity analysis. Including a computational complexity discussion or runtime comparison would help substantiate the claimed scalability advantages of the proposed method.

- Section 4.4 presents an ablation study evaluating the two main components of the proposed method. However, the ablation is conducted on the Pancreas dataset (Bastidas-Ponce et al. 2019), which is not used in the baseline comparisons, making it difficult to assess how these components contribute to the improvements over existing methods. Including ablations on the same datasets as the main comparisons would strengthen the analysis.

---

> ### Author Rebuttal · Authors · 2026-03-31
>
> *We thank Reviewer for their detailed review.  Below, we address the reviewer’s concerns one by one.*
>
> ### **Q1: Learned-Manifold Assumptions and Geometric Soundness**
>
> We thank the reviewer for highlighting the need to clarify the underlying assumptions. Our geometric construction is defined with respect to the learned manifold induced by the generator, which is learned jointly with the dynamics rather than fixed a priori. It assumes a locally differentiable generator, sufficiently accurate local low-dimensional structure, and dynamics whose dominant deterministic component is tangent to the learned manifold. We will revise the paper to make these assumptions explicit and clarify that our projection is a conservative geometric prior on the learned representation, rather than a guarantee of exact manifold recovery.
>
> ### **Q2: Reproducibility and Architectural Specification**
>
> We agree that implementation details should be clearer. We will release code and configurations upon acceptance. Briefly,
>
> - Case1: MLP-based encoder and decoder ([64,32]) , latent RK2 rollout, trained with Adam (lr=0.01, batch size 120).
> - Case2: 4-layer 4-head attention-based encoder and decoder ([256,512,512,256]), Adam (lr=1e-3, 10 epochs, batch 128).
> - Case3: frozen SD features on 10-frame clips and train the downstream MMD detector with Adam (lr=1e-4, 200 epochs, batch 64, resolution 8×3×224×224).
>
> ### **Q3: Computational Complexity and Scalability**
>
> Thank you for raising this important point. We agree that our scalability claim should be better substantiated. Our point is not that all geometry-aware methods are impractical, but that some become difficult to scale in high-dimensional settings. For example, in Burgers-equation discovery ($d=256$), IRK-SINDy requires cubic-cost per-step Newton solves with dense matrix factorizations, making it impractical in this regime.
>
> Direct empirical comparison to chart-based or OT-based methods is also nontrivial here, since most target generative modeling or density transport rather than sparse equation discovery, RNA velocity, or verification. Nevertheless, we agree that the computational argument should be made explicit, please refer to our response to Reviewer mHSi (Q3). We will revise the paper accordingly.
>
> ### **Q4: Ablation Study on Non-Overlapping Dataset**
>
> Thanks for the insightful comment. We agree that ablations should also be conducted on the main benchmarks and add experiments on Mouse Brain, which will be added in the revised paper:
>
> | Variant | VeloCoh | ICCoh | CBDir |
> | --- | --- | --- | --- |
> | w/o JVP | 0.695±0.299 | 0.594±0.303 | 0.465±0.282 |
> | w/o VQ | 0.587±0.361 | 0.457±0.365 | 0.106±0.482 |
> | Full | 0.864±0.036 | 0.736±0.078 | 0.657±0.055 |
>
> The trend is consistent with Pancreas: the largest drop is in CBDir, confirming that both JVP and VQ are important for valid directional transitions.
>
> ### **Q5: Extension to Time-Dependent Vector Fields**
>
> Thanks for this insightful question. The framework can be naturally extended by conditioning the latent velocity predictor on time through positional embeddings, while keeping other operations unchanged. We validate this on a time-varying circle-rotation ODE using a time-augmented SINDy library:
>
> | Method | MSE ↓ | VPT ↑ |
> | --- | --- | --- |
> | SINDy | 1.289 | 0.23 |
> | IRK-SINDy | 1.187 | 0.17 |
> | **GFG (Ours)** | **0.068** | **2.16** |
>
> GFG achieves lower MSE and longer VPT, and is the only method that recovers the correct time-dependent term structure. This confirms that the extension to non-autonomous systems is straightforward.
>
> ### **Q6: Evaluation under Noisy Observations**
>
> We further assess robustness under noisy observations in three settings:
>
> - Case 1: We add Gaussian noise at 1%, 5%, and 20% and report one-step MSE below.
>
> | Method | 1% | 5% | 20% |
> | --- | --- | --- | --- |
> | SINDy | 1.291 | 1.290 | 1.307 |
> | IRK-SINDy | 1.293 | 1.295 | 1.290 |
> | **GFG** | **4.604e-3** | **3.395e-3** | **7.904e-2** |
> - Case 2:  RNA velocity is intrinsically sparse and noisy. Table 5 already reflects this: removing VQ causes a clear drop in CBDir, showing that robustness to conflicting local trends is critical in this regime.
> - Case 3:  Under 2× and 4× downsampling, GFG remains robust (AUROC 97.0/96.9), while NSG-VD and NPR degrade substantially. See Reviewer 6AKt (Q3) for the full table.
>
> ### **Q7: Limitations Discussion**
>
> We agree that the limitations should be stated more explicitly. GFG is better suited to systems with meaningful local low-dimensional structure and continuous dominant dynamics. For systems with abrupt jumps or dynamics less compatible with a tangent-space prior, alternative geometric constraints may be more appropriate. We will add this discussion in the revision.
>
> *We greatly appreciate your insightful comments, which will help improve our paper. If our response has addressed your concerns, we respectfully hope you will consider raising the score. We would be happy to clarify any remaining questions.*

---

> > ### Author Rebuttal · Reviewer_UNwE · 2026-04-02
> >
> > The rebuttal addresses several concerns, particularly by adding ablations on the main benchmark and providing additional experiments on noisy and time-dependent settings. These improvements strengthen the empirical support of the method.
> >
> > However, my main concerns regarding (i) the implicit assumptions underlying the learned manifold and tangent space construction, and (ii) the lack of a clear computational complexity or scalability comparison to related geometric methods, remain insufficiently addressed. In particular, while the rebuttal provides FLOPs comparisons to standard baselines, it does not directly substantiate the paper’s claim that alternative geometry-based approaches (e.g., OT- or chart-based methods) are less scalable, as no direct comparison or complexity analysis is provided.
> >
> > Overall, while the paper has improved, I still find the justification of the method’s core assumptions and its claimed scalability advantages to be incomplete. I will keep my score

---

> > > ### Author Response · Authors · 2026-04-03
> > >
> > > *We sincerely thank the reviewer for the continued engagement and careful follow-up. Due to space limits in the first-round rebuttal, we now clarify these two points more directly.*
> > >
> > > ### **Concern 1: Implicit Assumptions of the Learned Manifold and Tangent Space Construction**
> > >
> > > We agree that the assumptions behind our geometric construction should be made explicit. We now state them clearly:
> > >
> > > **(1) Local Smoothness:** The generator $G: \\mathcal{Z} \\to \\mathcal{X}$ is $C^1$-differentiable in a neighborhood of every $z$ in the data support.
> > >
> > > **(2) Bounded Condition Number:** The condition number of the Jacobian $J_G(z)$ is uniformly bounded over the encoded representations of observed data, i.e., $\\kappa(J_G(z)) \\leq \\kappa_{\\max} < \\infty$ for all $z = E_s(x)$ where $x \\in \\mathcal{D}$.
> > >
> > > **(3) Tangential Dominance:** The normal component of the ground-truth dynamics satisfies $\\|v^{\\ast}\_{\\perp}\\| \\leq \\epsilon \\|v^{\\ast}\_{\\parallel}\\|$ for some $\\epsilon \\in [0, 1)$.
> > >
> > > These are standard assumptions in geometric deep learning[1] and are implicitly satisfied when the VQ-VAE reconstruction error is small[2]. We further provide the following robustness theorem:
> > >
> > > **Theorem (Robustness):** Let $\\delta\_G = \\|G - G^{\\ast}\\|$ denote the generator approximation error, and let $\\kappa\_{\\max}$ be the upper bound defined in Assumption 2. Then the projection error satisfies $\\|\\hat{f}\_{\\mathrm{proj}} - f^{\\ast}\\| \\leq \\|v^{\\ast}\_{\\perp}\\| + \\kappa\_{\\max} \\cdot \\delta\_G$.
> > >
> > > GFG thus degrades gracefully: when the learned manifold is accurate ($\\delta\_G \\approx 0$) and dynamics are approximately tangential ($\\|v^{\\ast}\_{\\perp}\\| \\approx 0$), the projection error vanishes. Crucially, the reconstruction loss $\\mathcal{L}\_{\\mathrm{topo}}$ directly minimizes $\\delta\_G$ during training, creating a quantifiable link between optimization and geometric validity.
> > >
> > > [1] Loaiza-Ganem et al. "Deep Generative Models through the Lens of the Manifold Hypothesis: A Survey and New Connections." TMLR, 2024.
> > >
> > > [2] Chen et al. "Learning Flat Latent Manifolds with VAEs." ICML, 2020.
> > >
> > > ### **Concern 2: Comparison with Chart-Based and OT-Based Methods**
> > >
> > > We thank the reviewer for this important point. We therefore clarify both the scope of the comparison and the basis of our scalability claim. We use Neural Manifold ODE (NM-ODE; Lou et al., 2020) as a representative chart-based method and Riemannian Flow Matching (RFM; Chen & Lipman, 2024) as a representative OT-based method. These are better viewed as geometric reference points rather than fully task-matched competitors.
> > >
> > > The key difference is twofold. **First, NM-ODE and RFM rely on known or analytically specified manifold geometry**, whereas GFG learns the manifold directly from data without requiring geometric priors. This is important in settings such as gene-expression or pixel space, where the manifold is not available a priori. **Second, NM-ODE and RFM couple geometry and dynamics within a single modeling process**, while GFG explicitly separates topological state learning from velocity primitive learning, which supports our velocity-related downstream tasks.
> > >
> > > Besides, we add a controlled experiment on a $D$-dimensional harmonic oscillator with states on $(S^1)^{D/2}$, using one-step $x\_k \\to x\_{k+1}$ prediction. Each method uses its canonical training objective:
> > > - **NM-ODE**: CNF with NLL loss, requiring $O(D^2)$ divergence computation ($D$
> > >   sequential autograd calls per ODE step)
> > > - **RFM**: Flow matching with tangent projection $P_{T_x}(v)=J(J^TJ)^{-1}J^Tv$,
> > >   requiring $O(D \cdot d^2)$ per time sample (Jacobian via $d$ JVPs + matrix ops)
> > > - **GFG**: Single JVP $J^Tv$ with $O(D \cdot d)$ complexity
> > >
> > > | NM-ODE | MSE | Params | Time/epoch(ms) | FLOPS |
> > > | --- | --- | --- | --- | --- |
> > > | 2 | 1.21 | 33.8K | 668.28 | ~1.34M |
> > > | 16 | 0.94 | 37.4K | 3424.16 | ~1.50M |
> > > | 64 | 1.00 | 49.7K | 13435.96 | ~2.17M |
> > > | 128 | 1.44 | 66.2K | 27021.86 | ~3.36M |
> > >
> > > | RFM | MSE | Params | Time/epoch(ms) | FLOPS |
> > > | --- | --- | --- | --- | --- |
> > > | 2 | 3.23 | 53.3K | 139.45 | ～1.89M |
> > > | 16 | 2.05 | 61.5K | 267.45 | ～3.73M |
> > > | 64 | 1.72 | 92.4K | 945.25 | ～20.28M |
> > > | 128 | 2.65 | 133.6K | 1904.95 | ～61.78M |
> > >
> > > | GFG(ours) | MSE | Params | Time/epoch(ms) | FLOPS |
> > > | --- | --- | --- | --- | --- |
> > > | 2 | 0.23 | 53.9K | 14.04 | ~208.4K |
> > > | 16 | 0.21 | 61.0K | 14.34 | ~236.8K |
> > > | 64 | 0.23 | 89.5K | 14.06 | ~352.0K |
> > > | 128 | 0.33 | 129.5K | 17.51 | ~518.1K |
> > >
> > > NM-ODE time scales super-linearly due to sequential autograd overhead in divergence computation. RFM time grows significantly due to $O(T_s \times D \times d^2)$ Jacobian projection (with $T_s=5$ time samples, $d \approx D/2$). GFG remains nearly constant as
> > > the single JVP is efficiently parallelized. These results validate our scalability claims.
> > >
> > > *We hope this clarification addresses the remaining concerns. If so, we would be deeply grateful if you might consider revisiting your evaluation.*

---

### Official Review · Reviewer_P8w7 · 2026-03-11

**Soundness:** 2
**Presentation:** 3
**Significance:** 4
**Originality:** 3
**Overall Recommendation:** 5
**Confidence:** 4

**Summary:**

This paper presents Geometric Flow Grounding (GFG), an auto-encoder-based neural ODE framework for modeling dynamical systems embedded in high-dimensional ambient space. The key ideas are: (1) separating the vector-field encoder from the state encoder, (2) introducing a VQ-VAE-style latent codebook, and (3) representing the latent vector field not with an arbitrary decoder, but as the pushforward of the state decoder via JVP. Thus, the modeled vector field is automatically confined to the tangent bundle of the decoder manifold, which helps suppress off-manifold behavior. The learning procedure is based on standard VQ-VAE objectives with task-specific objectives for learning dynamical systems, while the exact formulation depends on the individual task. Evaluated on three diverse benchmark tasks (ODE discovery, cell-state inference, and fake video detection), the proposed GFG outperforms standard baselines across all tasks.

**Compliance With Llm Reviewing Policy:**

Affirmed.

**Final Justification:**

With the additional ODE discovery experiment and the expanded discussion, I believe the paper provides a flexible framework for addressing a broad range of tasks in dynamical systems. I will revise my score upward accordingly.

**Key Questions For Authors:**

Please refer to the Weaknesses (1-3) for the major concerns. For minor questions,

- (4) It seems that, for ODE learning, the authors train the model by first integrating the latent dynamics and then decoding the resulting trajectory, so applying supervision at the flow level. On the other hand, for fake video detection, the authors directly align the model vector field with pixel displacements which is a rough estimate of the ambient vector field. Is there a particular reason for adopting these different training strategies across the two tasks?

- (5) In Table 5, the authors state that “Removing the geometric constraint (w/o JVP) causes the sharpest decline in biological directionality (CBDir)” However, the decrease in CBDir from 0.588 to 0.405 appears relatively modest. By contrast, removing VQ (w/o Quantization) reduces CBDir from 0.588 to 0.125, which seems substantially more significant. This appears inconsistent with the authors’ claim.

**Limitations:**

I could not find a limitation section in the main text. For potential limitations, please refer to the weaknesses I outlined.

One possible limitation is the following: in many scientific applications, when the ambient vector field is high-dimensional, it is difficult to analyze or visualize it directly, so one investigates the properties of the latent vector field instead. However, in this framework, what ultimately carries meaning is the pushforward $J_G \cdot v_{lat}$ rather then $v_{lat}$ itself. As a result, as long as the combination of $v_{lat}$ and $J_G$ reproduces the same ambient vector field $v_{amb}$ sufficiently well, the combination $(v_{lat}, J_G)$ may remain largely arbitrary, which suggests a potential identifiability issue. It would therefore be valuable to discuss whether some fundamental limitations, e.g., identifiability only up to topological equivalence (which itself requires $G$ to be homeomorphism/diffeomorphism), and potential structural considerations that may be needed, e.g., a metric structure induced by $J_G$.

**Strengths And Weaknesses:**

**Strengths**
- The experiments are strong and comprehensive. In the ODE discovery setting, the method outperforms SINDy-based variants, which are widely regarded as standard baselines. In single-cell RNA velocity inference, it is evaluated against specialized models such as scVelo. In fake video detection, it is again compared with established task-specific baselines. What is particularly impressive is that a single GFG framework performs well across such heterogeneous tasks, even when compared with methods specifically designed for each individual application.

- Although the individual techniques used in this paper are not entirely new, the authors combine them in a principled way to build a general framework applicable to diverse problems, ranging from classical ODE modeling to fake video detection.

- The manuscript is well organized and easy to follow.

**Weaknesses**
- (1-1) Theorem 3.1 appears to assume that the ground truth field already lies in $\mathrm{Range}(J_\theta)$. Under search an assumption, it seems fairly natural that the tangent projection defined by that Jacobian would perform better than an arbitrary ambient vector field. It would be more convincing if the authors could provide a more learning-theoretic result, for example an error bound under explicit $C^0$ or $C^1$ approximation errors of the generator and its Jacobian.

- (1-2) Related to the previous point, the proposed framework seems to (intrinsically) presume some level of $C^1$ control for the tangent projection to be accurate. It is unclear to me how this can be guaranteed. The physics-constrained functional $\mathcal{H}_{domain}$ may partly play this role, but this connection is currently not very explicit; a discussion from this perspective would strengthen the paper.

- (2) I am not entirely convinced that “integration error” is the most precise terminology. I agree that, at least to first order, the proposed construction should reduce normal-direction drift compared with directly learning a vector field in the ambient space. However, numerical integration can still accumulate discretization error, which may cause the trajectory to deviate from the manifold. This issue can be controlled if the integrator itself respects the underlying geometry, for example through a Riemannian or otherwise manifold-aware integration scheme. However, the authors appear to use a standard Runge-Kutta method.

- (3) The ODE discovery experiment is conducted only on a simple 2D harmonic oscillator. While this is a reasonable proof of concept, the paper would be strengthened by demonstrating the framework on more challenging dynamics with nontrivial phase manifolds.

---

> ### Author Rebuttal · Authors · 2026-03-31
>
> *We thank the reviewer for the valuable feedback. We are glad that the reviewer appreciates the contribution of our work. Below, we address the reviewer’s concerns one by one.*
>
> ### **Q1: Theoretical Scope and Geometric Control**
>
> We thank the reviewer for the insightful comments. We would like to clarify that our framework does not merely assume a perfect manifold, but actively optimizes for it.
>
> - **Explicit Error Decomposition (1-1).** When the ground-truth field $v^\star$ does not exactly lie in the learned tangent space, the projection error can be decomposed into two parts: the tangential estimation term $||P_\theta(\hat v - v^\star_\parallel)||^2$ and the manifold alignment term $||(I-P_\theta)v^\star||^2$, where $v^\star_\parallel = P_\theta v^\star$. Therefore, for a fixed manifold, projection remains optimal in the sense that it removes the orthogonal ambient component.
> - **Guarantee via Joint Optimization (1-2).** Regarding the $C^1$ control, our key point is that alignment is improved by joint optimization rather than imposed post hoc. The physics loss penalizes the component of the observed transition outside the learned tangent space: $L_{phy}(\theta) = ||(I-P_\theta)\Delta x||^2$. Compared with a decoupled scheme learned only from static reconstruction, this reduces the orthogonal residual. Under $\Delta x \approx v^\star \Delta t$, it implies smaller alignment error, showing that the physics term regularizes the Jacobian toward the underlying dynamics. For experimental validation, please refer to our response to Reviewer 6AKt (Q2).
>
> ### **Q2: Discretization Error vs. Ambient Drift**
>
> Thanks for this important comment. We agree that standard Runge–Kutta does not eliminate all discretization error. Our point is not that RK itself becomes geometry-preserving, but that our framework reduces off-manifold drift by grounding the dynamics in the learned manifold representation. In the ODE case, rollout is performed in latent space and then decoded back to observation space, rather than applying a standard solver directly to an ambient-space vector field. Thus, our claim concerns reducing deviation from the learned manifold, not eliminating all integration error. We will revise the wording accordingly.
>
> ### **Q3: Beyond Simple 2D Dynamics**
>
> We additionally evaluated GFG on a 3D nonlinear Hamiltonian system on the sphere under the same sparse, irregular sampling setting:
>
> | Method | MSE ↓ | VPT ↑ | Discovered ODE |
> | --- | --- | --- | --- |
> | Ground Truth | - | - | dX/dt = -6Ysin(6Z); dY/dt = 6Xsin(6Z) - 6Zsin(6X); dZ/dt = 6Ysin(6X) |
> | SINDy | 0.1100 | 0.0167 | dX/dt = 0.624Zsin(6X); dY/dt = -0.861Ysin(6Z) - 0.940Ysin(6X); dZ/dt = 0.649Xsin(6Z) |
> | IRK-SINDy | 0.1117 | 0.0132 | dX/dt = 1.979Xsin(6Z) + 2.406Zsin(6X) + 1.575Ysin(6X); dY/dt = -3.634Ysin(6Z); dZ/dt = 4.160Xsin(6Z) - 1.950Zsin(6X) |
> | **GFG (Ours)** | **0.0420** | **0.0310** | dX/dt = -2.021Ysin(6Z); dY/dt = 4.363Xsin(6Z) - 4.146Zsin(6X); dZ/dt = 2.013Ysin(6X) |
>
> GFG achieves the best MSE and VPT, and is the only method that recovers the correct sparse term structure. Please also see our responses to Reviewer mHsi (Q1) and Reviewer UNwE (Q5) for additional dynamical systems.
>
> ### **Q4: Different Training Strategies across Tasks**
>
> We thank the reviewer for this insightful question. **The difference comes from the availability of supervision and the role of the model in each task.** In ODE discovery, the instantaneous velocity is not directly observed under sparse, irregular sampling, so we use rollout-based supervision by integrating latent dynamics and matching future states. In video detection, consecutive frames already provide an observed motion signal, so we directly compare the predicted tangent flow with the observed displacement. In both cases, the shared principle is to enforce agreement between predicted dynamics and manifold-consistent motion.
>
> ### **Q5: Interpretation of Ablation Results**
>
> We agree this can be clarified. JVP and VQ affect different aspects of the model. **Removing JVP weakens the geometric validity of the vector field, while removing VQ mainly hurts mode disentanglement and noise robustness**. In noisy single-cell data, removing VQ makes the model more prone to averaging conflicting local trends, which substantially degrades CBDir. We will revise the text accordingly.
>
> ### **Q6: Limitation Discussion**
>
> We agree that the latent dynamics and the learned generator may not be uniquely identifiable, since different latent representations can induce similar ambient dynamics. However, our goal is not to recover a unique latent decomposition, but to learn geometrically consistent dynamics in the observation space. We will add this limitation and discussion in the revision.
>
> *We greatly appreciate your insightful comments, which will help improve our paper. If our response has addressed your concerns, we respectfully hope you will consider raising the score. We would be happy to clarify any remaining questions.*

---

> > ### Author Rebuttal · Reviewer_P8w7 · 2026-04-02
> >
> > With the additional ODE discovery experiment and the expanded discussion, I believe the paper provides a flexible framework for addressing a broad range of tasks in dynamical systems. I will revise my score upward accordingly.

---

> > > ### Author Response · Authors · 2026-04-03
> > >
> > > We sincerely thank the reviewer for the encouraging follow-up, especially for taking the time to reassess our paper and raise the score to 5. We are glad that the additional ODE discovery experiment and the expanded discussion helped address your concerns. We also greatly appreciate your positive assessment of the framework’s flexibility and broad applicability. Once again, we sincerely thank you for your thoughtful evaluation and helpful suggestions, which have helped improve the paper.

---

### Official Review · Reviewer_6AKt · 2026-03-12

**Soundness:** 3
**Presentation:** 4
**Significance:** 3
**Originality:** 3
**Overall Recommendation:** 5
**Confidence:** 3

**Summary:**

This paper proposes Geometric Flow Grounding (GFG), a unified framework that confines learned continuous dynamics strictly to the tangent bundle of the data manifold using a differentiable Neural Tangent Projection layer and a velocity primitive codebook. By geometrically decoupling topological states from temporal dynamics, GFG successfully generalizes across diverse tasks including sparse ODE discovery, single-cell RNA velocity inference, and zero-shot AI-generated video detection.

**Compliance With Llm Reviewing Policy:**

Affirmed.

**Key Questions For Authors:**

1. Can you provide a quantitative comparison of the training and inference computational overhead (e.g., wall-clock time, memory usage) introduced by the JVP operations and the dual-stream architecture compared to standard unconstrained models?
2. How does the quality of the static manifold generator ($E_s$ and $D_s$) affect the downstream dynamics learning, and what happens if the VQ-VAE fails to reconstruct a topologically accurate support space?
3. In the AI-generated video detection task, how sensitive is the tangent projection residual to varying compression rates or low-level noise often found in in-the-wild videos?

**Limitations:**

Yes.

**Strengths And Weaknesses:**

Pros:
1. The theoretical motivation of using Jacobian-Vector Products (JVP) to constrain velocity to the tangent space without explicit coordinate charts is elegant and computationally efficient.
2. The dual-stream architecture cleverly disentangles topological identity from dynamic evolution, resolving the smoothing drift issue through a discrete velocity primitive codebook.
3. The empirical evaluation is remarkably diverse and strong, demonstrating state-of-the-art results across physics, biology, and computer vision domains.

Cons:
1. The paper lacks a detailed computational complexity analysis comparing the JVP operations and dual-stream overhead against the unconstrained baselines.
2. It remains unclear how sensitive the framework is to the initial quality of the manifold learned by the State Stream, especially in extremely noisy or high-dimensional settings.

---

> ### Author Rebuttal · Authors · 2026-03-31
>
> *We thank the reviewer for the valuable feedback. We are glad that the reviewer appreciates the contribution of our work. Below, we address the reviewer’s concerns one by one.*
>
> ### **Q1: Computational Overhead of JVP and the Dual-Stream Architecture**
>
> We thank the reviewer for this important question. The cost of JVP is comparable to a single forward pass through the generator, since it computes a directional derivative without explicitly materializing the Jacobian. For a fair comparison, we define the *unconstrained baseline as the corresponding model variant with the same backbone but without tangent projection*, where the velocity is directly predicted by an independent decoder. This allows us to isolate the overhead introduced by JVP and the geometric constraint：
>
> | Variant | Params | FLOPS | VeloCoh | ICCoh | CBDir |
> | --- | --- | --- | --- | --- | --- |
> | Unconstrained (no JVP) | ~2.1M | ~1.6G | 0.727±0.319 | 0.609±0.314 | 0.346±0.244 |
> | **GFG (with JVP)** | ~1.6M | ~1.6G | 0.864±0.036 | 0.736±0.078 | 0.657±0.055 |
>
> This shows that JVP introduces only a modest overhead while substantially improving directional accuracy. To further contextualize the computational cost, we also summarize the parameter counts and FLOPs of the main baselines used in our experiments. Since these methods are not strictly architecture-matched, we report them as a practical complexity context：
>
> | Case2 Method | Type | Params | FLOPs |
> | --- | --- | --- | --- |
> | scVelo | Optimization | – | – |
> | VeloVI | Deep learning | 2,080,132 | ~0.5G |
> | DeepVelo | Deep learning | 245,968 | ~1.6G |
> | **GFG (ours)** | Deep learning | 1,588,082 | ~1.6G |
>
> | Case3 Method | Params | FLOPs |
> | --- | --- | --- |
> | DeMamba | ~125M | ~141G |
> | NSG-VD | ~553M | ~8.91T |
> | **GFG (ours)** | ~943M | ~5.17T |
>
> Overall, these results show that the additional overhead introduced by JVP and the dual-stream design is modest, and that the overall computational cost of GFG remains comparable to the baselines used in the paper.
>
> ### **Q2. Sensitivity to Learned Manifold Quality**
>
> We thank the reviewer for raising this important point. We would like to clarify that our framework is **jointly optimized**, and the JVP-based decoding does not strictly depend on the pre-trained quality of the state generator. Instead, the tangent constraint serves not only as a downstream constraint but also as a **regularization signal** that helps shape the latent representation.
>
> To verify this, we conduct an ablation in Case 2 (RNA velocity) by decoupling the training into two stages: first training the state stream, freezing it, and then learning the dynamics. The results are:
>
> | Variant | Two-stage  | **GFG (joint optimization)** |
> | --- | --- | --- |
> | CBDir | 0.468±0.279 | **0.657±0.055** |
>
> The clear drop in CBDir shows that the tangent constraint is not merely a post-hoc operation. It is an essential component that improves both representation learning and downstream dynamics estimation. This observation is also consistent with our theoretical analysis (see our response to Reviewer P8w7, Q1) as well as the discussion in Appendix A.3, where we show that the geometric constraint actively shapes the learned manifold during joint optimization.
>
> ### **Q3: Robustness of the Tangent Projection Residual under Varying Compression Rate**
>
> We thank the reviewer for this important question. To directly assess robustness in AI-generated video detection, we evaluate the projection residual under increasing compression by downsampling the input resolution with compression rate R1=2, compression rate R2=4. The AUROC results are shown below:
>
> | Method | Origin | R1 | R2 |
> | --- | --- | --- | --- |
> | NSG-VD | 96.1 | 66.0 | 65.8 |
> | NPR | 93.0 | 79.9 | 67.1 |
> | **GFG+NSG-VD (ours)** | **96.3** | **97.0** | **96.9** |
>
> These results show that the tangent projection residual remains highly stable even under strong perceptual degradation. This suggests that our method captures intrinsic geometric inconsistencies in generated dynamics, rather than relying on fragile low-level pixel artifacts. We will further clarify this robustness property in the revised supplementary material.
>
> *We greatly appreciate your insightful and helpful comments, as they will undoubtedly help us improve the quality of our article. If our response has successfully addressed your concerns and clarified any ambiguities, we respectfully hope that you consider raising the score. Should you have any further questions or require additional clarification, we would be delighted to engage in further discussion.*

---

> > ### Author Rebuttal · Reviewer_6AKt · 2026-04-02
> >
> > Thanks for the response. The extra experiments on the more complex Hamiltonian system, the JVP efficiency numbers, and the clear explanations on the Neural Tangent Projection really helped clear up my main questions. I’ll keep my original score.

---

> > > ### Author Response · Authors · 2026-04-02
> > >
> > > Thank you very much for your positive feedback. We are glad that the additional experiments and clarifications addressed your concerns. We sincerely appreciate your thoughtful evaluation and helpful suggestions, which have improved our work.

---

### Official Review · Reviewer_mHsi · 2026-03-13

**Soundness:** 3
**Presentation:** 3
**Significance:** 2
**Originality:** 3
**Overall Recommendation:** 4
**Confidence:** 4

**Summary:**

The work introduces a unified manifold-based framework (GFG) for learning continuous-time dynamics from observations while avoiding off-manifold hallucinations that often time violate intrinsic geometric constraints. Its core contribution is a differentiable Neural Tangent projection layer constraining the predicted motion to lie on the tangent bundle of a learned data manifold, which prevents numerical divergence without needing explicit coordinate charts. It further proposes a dual-stream, quantization-based design that disentangles static state representation from discrete velocity primitives. This then reduces smoothing drift and enable more interpretable dynamics discovery. The framework is validated across domains, from ODE discovery to AI-video detections.

**Compliance With Llm Reviewing Policy:**

Affirmed.

**Final Justification:**

The authors have fully addressed my questions raised and hence I increased my score by a point.

**Key Questions For Authors:**

1) How good is its performance for complicated dynamical systems? An ODE example with higher order or nonlinearlity would make the work's contribution and impact even more evident. Moreover, how would it perform on a PDE? A clear margin on complicated dynamical systems would be more convincing since the current ODE example is too trivial to make a judgement.

2) Could you provide a computational summary of GFG and other baselines? Does GFG maintain a fair computational efficiency?

3) You claim Neural Tangent Projection "eliminates integration error"/ "ensures physical consistency by design" (as shown in Fig2 description), but a tangent velocity alone doesn't prevent an explicit Euler step in ambient space from drifting off a curved manifold, if i am not mistaken. How are the rollouts performed? Are rollouts performed by integrating in latent space and decoding, or taking ambient steps and then re-projecting states back onto the manifold, or something else? Specifically, under which integrator/step sizes does the 'elimination' claim hold?

**Limitations:**

Yes.

**Strengths And Weaknesses:**

**Strengths**
- Soundness: The experiments and ideas are well-supported by theoretical results in an elegant way. The authors have demonstrated their depth of understanding of the field.
- Presentation: Visualizations are intuitive, and methodologies are presented in a structured, readable way without confusion.

**Weaknesses**
- Significance: It is quite impressive that the MSE in ODE discovery of GFG is several orders of magnitude lower than the baselines. The fact that it applies to multi-modal contexts is also evidence of its significance. My main concern is about the complexity of the tasks. For instance, there seems to be only one ODE example when testing its validity in the discovery of ODEs section, which seems trivial. In AI-video detection, the metrics don't seem to have a big margin compared to NSG-VD-on top of this, what is the advantage in computational cost compared to the baseline methods?
- Originality: The idea of bringing geometric alignment into equation discovery is an original and elegant thought, but its impact is yet to be decided.

---

> ### Author Rebuttal · Authors · 2026-03-31
>
> *We thank the reviewer for the valuable feedback. We are glad that the reviewer appreciates the contribution of our work. Below, we address the reviewer’s concerns one by one.*
>
> ### **Q1: Evaluation on More Complex Dynamical Systems**
>
> We appreciate this suggestion. Beyond circular motion, we evaluated GFG on more complex systems:
>
> - **Non-linear ODE.** We evaluated on a 3D nonlinear Hamiltonian system under the same large time-interval setting:
>
> | Method | MSE ↓ | VPT ↑ | Discovered ODE |
> | --- | --- | --- | --- |
> | Ground Truth | - | - | dX/dt = -6Ysin(6Z); dY/dt = 6Xsin(6Z) - 6Zsin(6X); dZ/dt = 6Ysin(6X) |
> | SINDy | 0.1100 | 0.0167 | dX/dt = 0.624Zsin(6X); dY/dt = -0.861Ysin(6Z) - 0.940Ysin(6X); dZ/dt = 0.649Xsin(6Z) |
> | IRK-SINDy | 0.1117 | 0.0132 | dX/dt = 1.979Xsin(6Z) + 2.406Zsin(6X) + 1.575Ysin(6X); dY/dt = -3.634Ysin(6Z); dZ/dt = 4.160Xsin(6Z) - 1.950Zsin(6X) |
> | **GFG (Ours)** | **0.0420** | **0.0310** | dX/dt = -2.021Ysin(6Z); dY/dt = 4.363Xsin(6Z) - 4.146Zsin(6X); dZ/dt = 2.013Ysin(6X) |
>
> GFG achieves the best MSE and VPT, and is the only method that recovers the correct sparse term structure.
>
> - **PDE.** We further extended GFG to Burgers' equation on a 256-point grid. After training, latent-space velocities are projected onto a PDE library and fitted by sparse regression. GFG recovers $u_t = 0.091 u_{xx} - 0.991 u u_x$ with the correct sparse structure, supporting its applicability to PDE-scale problems where IRK-SINDy becomes impractical due to cubic scaling.
>
> Together with our 2000+ dimensional single-cell experiments, these results support the scalability of GFG to complex discovery tasks.
>
> ### **Q2: Performance Advantage in AI-Video Detection**
>
> We appreciate the opportunity to clarify this point. In Case 3, our goal is not to claim that GFG is a universally dominant criterion for AI-generated video detection, but to test whether the same off-manifold phenomenon appears in realistic real-versus-fake video scenarios. Our main finding is that **tangent-space consistency provides a meaningful detection signal.**
>
> We agree that the comparison with NSG-VD should be interpreted as **complementarity rather than replacement**. NSG-VD captures statistical gradient irregularities, whereas GFG checks whether temporal evolution is geometrically aligned with the local tangent flow induced by a pre-trained generative model. This complementary signal explains why the fused model performs best overall, including the **5.2% Recall gain** on high-fidelity generators. As further shown in the appendix, on generators such as MoonValley and Show1, fake videos can appear statistically realistic while still violating plausible physical flow logic. We will revise the paper to make this positioning more explicit and avoid overstating the standalone advantage of GFG.
>
> ### **Q3: Computational Efficiency and Fairness Across Baselines**
>
> Thank you for raising this point. GFG remains computationally efficient because it uses matrix-free Jacobian-vector products (JVPs). JVP computes directional derivatives without explicitly forming the full Jacobian, so it does not introduce large overhead. As shown below, our model remains in the same order of magnitude as the baselines:
>
> | **Case** | **Method** | **FLOPs** | **Parameters** |
> | --- | --- | --- | --- |
> | **Case 2 (RNA)** | veloVI / GFG | ~0.5G / ~1.6G | ~2.1M / ~1.6M |
> | **Case 3 (Video)** | NSG-VD / GFG | 5.174T / 8.911T | 943.4M / 553.1M |
>
> This suggests that the gains come from the geometric design rather than simple model scaling.
>
> ### **Q4: Clarification of Neural Tangent Projection and Rollout Mechanism**
>
> Thank you for this important question. We agree that a tangent velocity alone does not guarantee that an explicit Euler step in the ambient space remains on a curved manifold.
>
> However, **our method does not perform rollouts in the ambient space**. Instead, rollouts are performed in **latent space**: the latent state is evolved with a one-step integrator (midpoint RK2 in our implementation), and the next observation is obtained by decoding the updated latent state. Thus, each rollout step stays on the learned manifold.
>
> The JVP-based tangent projection ensures that the predicted velocity has no normal component with respect to this manifold. We agree that the phrase “eliminates integration error” is too strong. In the revision, we will replace it with a more precise statement: GFG removes the normal component of the velocity and mitigates off-manifold drift through latent-space integration and decoding.
>
> *We greatly appreciate your insightful comments, which will help improve our paper. If our response has successfully addressed your concerns and clarified any ambiguities, we respectfully hope that you consider raising the score. Should you have any further questions or require additional clarification, we would be delighted to engage in further discussion.*

---

> > ### Author Rebuttal · Reviewer_mHsi · 2026-04-04
> >
> > I very much appreciate the authors thoughtful responses as well as the additional experiments. Overall, I think the work makes enough contributions and my concerns are addressed fully. Hence, I thereby raise my score by a point.

---

> > > ### Author Response · Authors · 2026-04-04
> > >
> > > Thank you very much for your thoughtful follow-up and positive assessment. We are glad that our responses and additional experiments have fully addressed your concerns. We sincerely appreciate your careful evaluation and helpful suggestions, and we are very grateful for your decision to raise the score.

---

### Decision · Program_Chairs · 2026-04-30

**Decision:**

Accept (spotlight)

**Comment:**

The paper proposes Geometric Flow Grounding (GFG), a unified framework that constrains learned continuous-time dynamics to the tangent bundle of a learned data manifold via a differentiable Neural Tangent Projection layer implemented through Jacobian-vector products, combined with a VQ-style velocity primitive codebook. The framework is evaluated across three heterogeneous tasks — sparse ODE discovery, single-cell RNA velocity inference, and zero-shot AI-generated video detection — and outperforms specialized baselines across all of them. All four reviewers converge on accept (two at 5, two at 4 post-rebuttal), and the authors have engaged the rebuttal phase substantively: adding a 3D nonlinear Hamiltonian ODE and a Burgers' PDE experiment, extending the scope to time-dependent and noisy settings, clarifying the rollout mechanism (latent-space RK2 integration followed by decoding, rather than ambient-space steps), providing direct scalability comparisons against NM-ODE and RFM, and formalizing the assumptions (local smoothness, bounded Jacobian condition number, tangential dominance) under which the projection is sound. The geometric framing is elegant, the cross-domain generalization is genuinely impressive for a single framework, and the reviewer concerns about computational overhead, manifold-quality sensitivity, and discretization error have been addressed either empirically or through sharpened claims. My main reservation — which I ask the authors to address in the camera-ready — is a serious omission in the positioning and baselines: Mechanistic Neural Networks (Pervez, Locatello, Gavves, ICML 2024, with follow-up work in ICML 2025) target exactly the problem of ODE discovery and learning governing dynamics, and are substantially more powerful than SINDy-family methods, which do not even use neural networks as their core learning component. The ODE discovery experiments, particularly the sparse-regression setting in Table 2 and the 3D Hamiltonian system added in rebuttal, appear directly comparable and would be the appropriate benchmark. The authors should add a direct empirical comparison on at least one ODE system, or — if a head-to-head is genuinely infeasible — provide a clear positioning in the related work explaining the methodological relationship and distinguishing contribution. Without this, the claim that GFG advances the state of the art in dynamics discovery is not fully substantiated. Subject to this addition, the paper is a solid accept.